# Space-time landslide hazard modeling via Ensemble Neural Networks

Ashok Dahal[1], Hakan Tanyas[1], Cees van Westen[1], Mark van der Meijde[1], Cees van Westen[1], P. Martin Mai[2], Raphael Huser[3], and Luigi Lombardo[2]

[1]University of Twente, Faculty of Geo-Information Science and Earth Observation (ITC), PO Box 217, Enschede, AE 7500, Netherlands
[2]Physical Science and Engineering (PSE) Division,, King Abdullah University of Science and Technology (KAUST), Thuwal 23955-6900, Saudi Arabia
[3]Statistics Program, Computer, Electrical and Mathematical Sciences and Engineering (CEMSE) Division,, King Abdullah University of Science and Technology (KAUST), Thuwal 23955-6900, Saudi Arabia

**Correspondence:** Ashok Dahal (a.dahal@utwente.nl)

**Abstract.** Until now, a full numerical description of the spatio-temporal dynamics of a landslide could be achieved only via physically-based models. The part of the geoscientific community developing data-driven model has instead focused on predicting where landslides may occur via susceptibility models. Moreover, they have estimated when landslides may occur via models that belong to the early-warning-system or to the rainfall-threshold themes. In this context, few published researches have explored a joint spatio-temporal model structure. Furthermore, the third element completing the hazard definition, i.e., the landslide size (i.e., areas or volumes), has hardly ever been modeled over space and time. However, technological advancements in data-driven models have reached a level of maturity that allows to model all three components (Where, When and Size). This work takes this direction and proposes for the first time a solution to the assessment of landslide hazard in a given area by jointly modeling landslide occurrences and their associated areal density per mapping unit, in space and time. To achieve this, we used a spatio-temporal landslide database generated for the Nepalese region affected by the Gorkha earthquake. The model relies on a deep-learning architecture trained using an Ensemble Neural Network, where the landslide occurrences and densities are aggregated over a squared mapping unit of $1 \times 1$ km and classified/regressed against a nested 30 m lattice. At the nested level, we have expressed predisposing and triggering factors. As for the temporal units, we have used an approximately 6-month resolution. The results are promising as our model performs satisfactorily both in the susceptibility (AUC = 0.93) and density prediction (Pearson r = 0.93) tasks over the entire spatio-temporal domain. This model takes a significant distance from the common susceptibility literature, proposing an integrated framework for hazard modelling in a data-driven context.

To promote reproducibility and repeatability of the analyses in this work, we share data and codes in a github repository accessible from this link.

## 1 Introduction

The literature on physically-based models for landslides shows various solutions of how to estimate where landslides can occur, when they occur, and how they may evolve (e.g., Formetta et al., 2016; Bout et al., 2018). This framework allows one

to describe the dynamics of a landslide from its initiation, propagation, and entrainment to the runout and deposition (e.g., Burton and Bathurst, 1998; Zhang et al., 2013). As a result, metrics such as the velocity, runout height, overall landslide area, and volume constitute standard outputs of such a modeling approach (see, van den Bout et al., 2021b, a). However, these models are often constrained to single slopes or catchments because of the spatial data requirements on geotechnical parameters. This limitation has stimulated the geoscientific community to develop data-driven models instead (Van Westen et al., 2006). These are much more versed to be extended over large regions because, rather than requiring specific geotechnical properties, they can rely on terrain attributes and remotely sensed data acting as geotechnical proxies (Van Westen et al., 2008; Frattini et al., 2010). However, in doing so, the geoscientific community has almost exclusively focused on assessing where landslides may occur, as temporal landslide data was hardly available. This notion is commonly referred to as landslide susceptibility (Reichenbach et al., 2018; Titti et al., 2021). As for the lesser number of publications focused on estimating when or how frequently landslides may occur at a given location, the community has produced a number of near-real-time predictive landslide models for rainfall (Intrieri et al., 2012; Kirschbaum and Stanley, 2018; Ju et al., 2020) and seismic (Tanyaş et al., 2018; Nowicki Jessee et al., 2018) triggers. With regard to characteristics such as velocity, kinetic energy and runout, albeit fundamental to describe a potential landslide threat (Fell et al., 2008; Corominas et al., 2014), these are currently impossible to be data-driven-modeled because no observed dataset of landslide dynamics exists to support the modelling and predicting paradigm of an Artificial Intelligence (AI). Guzzetti et al. (1999) proposed to alternatively model landslide areas, which can be easily extracted from a polygonal inventory. Nevertheless, the first spatially-explicit models able to estimate landslide areas have been recently proposed by Lombardo et al. (2021); Zapata et al. (2023). In their work, the authors exclusively estimated the potential landslide size at a given location without informing whether the given location would have been susceptible in the first place. This limitation has been further addressed by Bryce et al. (2022) and Aguilera et al. (2022), implementing models that couple susceptibility and landslide area prediction together. Nevertheless, even in these cases, the absence of the temporal dimension in their work implies that no current data-driven model is capable of solving the landslide hazard definition (Guzzetti et al., 1999), jointly estimating where, when (or how frequently) and how large landslides may be in a given spatio-temporal domain. Apart from spatial modelling, temporal aspects of landslides are also addressed in works of Samia et al. (2020); Ozturk et al. (2021).

The present work expands on the data-driven literature summarized above by proposing a space-time deep-learning model based on an Ensemble Neural Network (ENN) architecture. Neural Networks (NN) are not new to the landslide literature, although they have found the spotlight so far mostly for automated landslide detection (Catani, 2021; Meena et al., 2022), monitoring (Neaupane and Achet, 2004; Wang et al., 2005) and for landslide susceptibility assessment (Lee et al., 2004; Catani et al., 2005; Gomez and Kavzoglu, 2005; Grelle et al., 2014; Montrasio et al., 2014; Catani et al., 2016; Nocentini et al., 2023). Here, the main difference is that our ENN is built as an ensemble made of two elements, i.e., a landslide susceptibility classifier and a landslide density area regression model, both simultaneously defined over the same spatio-temporal domain. Thanks to the open data repository of Kincey et al. (2021), we tested our space-time ENN complying for the first time with the landslide hazard definition (as per Guzzetti et al., 1999).

The manuscript is organized as follows: Section 2 describes the data we used; Section 4 summarizes how we partitioned the study area; Section 5 lists the predictors we chose; Section 6 details our space-time ENN architecture; Section 7 reports our results, which are then discussed in Section 8, and Section 9 concludes our contribution with an overall summary and future plans.

## 2 Study area and landslide inventory

The 2015 Gorkha (Nepal) Earthquake is one of the strongest recent earthquakes in South Asia, and specifically along the Himalayan sector (e.g., Kargel et al., 2016). The Mw 7.8 mainshock occurred on $25^{th}$ April 2015, and together with a sequence of aftershocks, it was responsible for triggering more than 25,000 landslides (Roback et al., 2018). The ground motion did not only affect the Nepalese terrain right after the earthquake through co-seismic landslides, but its disturbance increased the landslide susceptibility in the following years, a phenomenon commonly referred to as earthquake legacy (Jones et al., 2021; Tanyaş et al., 2021a). The legacy of the Gorkha earthquake has been recently demonstrated by mapping a multi-temporal inventory, which has been publicly shared by Kincey et al. (2021). The authors mapped landslides across the area shown in Figure 1 from 2014 to 2018, including the co-seismic phase, as well as all pre-monsoons and post-monsoons seasons, with an approximate temporal coverage of six months. They used a time series of freely available medium-resolution satellite imagery (Landsat-8 and Sentinel-2) and aggregated the resulting landslide areas at the level of a 1 km squared lattice. Overall, they mapped three pre-seismic and seven post-seismic landslide inventories in addition to the co-seismic one. In this work, we excluded three pre-seismic inventories and selected the inventories from April 2015 onward because the effect of the ground motion and its legacy effect is present only after the event. As a result, from the gridded database by Kincey et al. (2021), we extracted a total of eight landslide inventories.

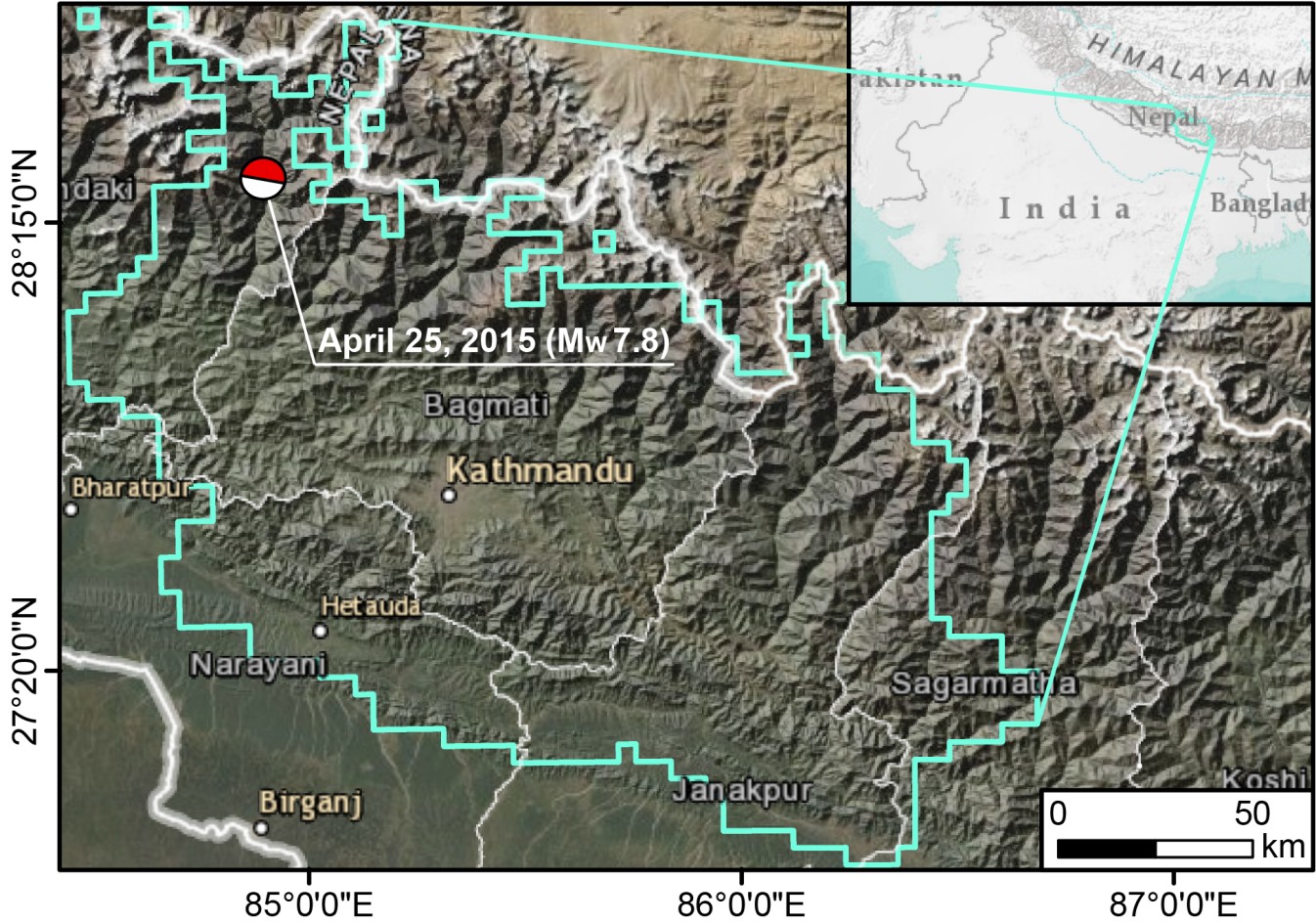

**Figure 1.** Study area defined within the cyan polygon, where Kincey et al. (2021) mapped the multitemporal landslide inventories upon which we based the analysis in this work. The Beach Ball shows the moment tensor of the energy release from 2015 Gorkha Earthquake.

Since the landslide information was aggregated at a 1 km resolution, it is not possible to disentangle single landslides, one from the others. Thus, each 1 km grid reports the whole landslide area mapped by the authors each time without excluding the footprint of previous failures. For this reason, we had to include a pre-processing step where each temporal replicate was re-calculated and re-assigned with the difference in landslide area density between two original subsequent inventories. In the attempt to focus on newly activated landslides, we have then considered only grid cells with an increase in landslide area. The
interpretation here is that an increase with time implies either newly formed landslides or re-activated ones. Conversely, the grids where the landslide area diminished with respect to their previous counterpart were assigned with a zero value under the assumption that no landslide took place, but rather vegetation recovery was responsible for the estimated change. The resulting temporal inventory at different time periods over the 1 km grid is shown in Figure 2.

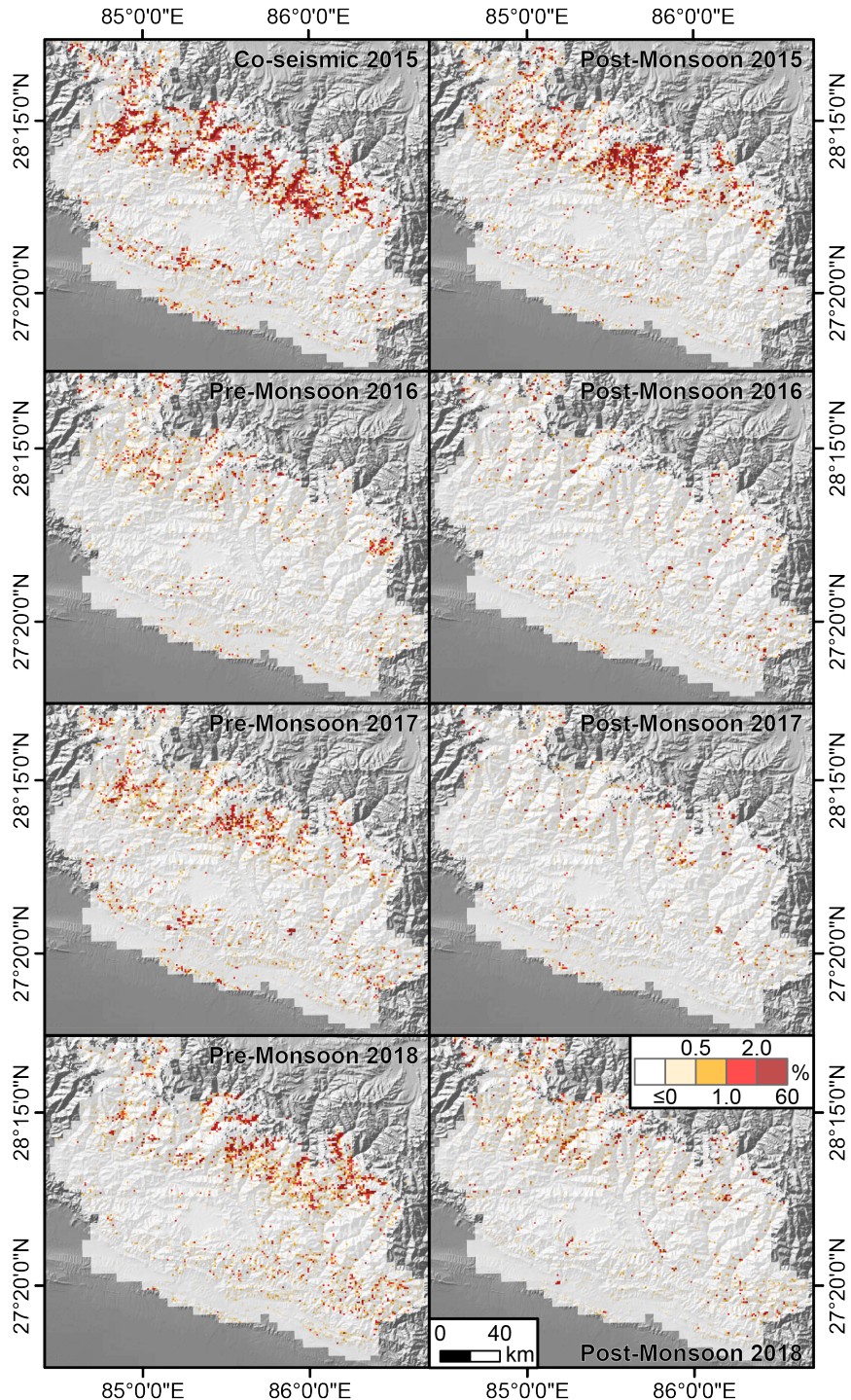

**Figure 2.** Landslide Area Density (% in a 1 km² grid) calculated as the difference between two consecutive inventories mapped with different time ranges provided by Kincey et al. (2021). The colorbar is saturated between 2 and 60 % because there are very few grid cells with such landslide area density in the data.

## 3 Geological context

Geology largely controls the landslide initiation and flow, and thus, it is commonly used as part of landslide susceptibility studies (Fan et al., 2019). In the context of this experiment, the majority of the study area (9%) is classified as Siwalik formations, followed by the Himal group and a combination of many river formations such as *Seti* and *Sarung Khola* formations (Dahal, 2012). The Siwalik Formation is mostly a Molasse deposit of the Himalayas, consisting of sandstone, mudstone, shale and conglomerate. The river formations that predominantly outcrop in the middle Himalayas are a combination of Schist, Granite, Gneiss, Phyllite, and Quartzite. The upper Himalayan region consists instead of a combination of Schist, Gneiss, Migmatites, and Marbles (Upreti, 2001).

This general overview is summarized in the geological map by Dahal (2012). However, this map does not cover a substantial portion of the upper Himalayas, where the landslide multi-temporal inventory mapped by Kincey et al. (2021) includes many landslides. Therefore, this map is largely unsuitable for testing a data-driven model aimed at explaining the landslide hazard associated with the abovementioned inventory.

A second geological map covers the Nepalese territory, this being made by Dahal (2012). However, it is limited to a very coarse resolution (1:1,000,000) and only reports information about the formations rather than the associated material. Even if one would extrapolate the respective rock types from the literature, (Upreti, 2001), such formations only summarize the complexity of the Himalayan landscape in a few classes, thus making it of limited use in the context of data-driven landslide modeling. The Department of Mines and Geology of Nepal is currently collating all the information with the intent of producing a detailed geological map at a 1:50,000 scale. However, this is a work in progress that still misses most of the study areas considered in this study.

This majors limitations affected our ability to consider geology in this contribution. Therefore, albeit we are aware of its relevance in any landslide study, geological information will not be explicitly included as a predictor in the modeling protocol presented below.

## 4 Selection of mapping units

To partition our study area, we use the same mapping unit defined by Kincey et al. (2021). Because the authors aggregated the landslide information on a $1 \times 1$ km$^2$ square grid, our model targets are defined within the same lattice structure. As for the definition of the predictor set, unlike current data-driven practices where medium resolution mapping units are assigned with the mean and standard deviation of the predictors under consideration (Ardizzone et al., 2002; Schlögel et al., 2018), here we exploit the NN structure to treat each predictor as an image. In other words, each $1 \times 1$ km$^2$ square grid was not summarized with its mean and standard deviation values, but rather, we provided the entire spatial distribution of predictors as an image patch to the CNN model, which is capable of reading image data.

Only feeding a single grid structure to the NN would neglect any spatial dependence coming from neighboring areas (Glenn et al., 2006; Vasiliev, 2020). Since landslides are dynamic phenomena, it is essential to inform the model about how the landslide distribution changes across the neighboring landscape, as well as the characteristics of the neighborhood under consideration. To do so, we extended the spatial vision of our ENN by creating two additional sets of lattices, each encompassing

sixteen 1 km grids, in a $4 \times 4$ patch. Figure 3 further explains the mapping unit structures, wherein in panel (a), we can observe that the 1 km red polygonal lattice created by Kincey et al. (2021) contains $32 \times 32$ pixels of the underlying terrain characteristics.

The subplot (b) shows how each patch is generated through the green boxes containing 16 inventory grids. Each box will later be used as the training patches in the ENN, which in turn implies a $128 \times 128$ pixels structure (32 pixels $\times$ 4 = 128) as input data. The model will then output 16 inventory grids, following the same data structure expressed at the $4 \times 4$ patch level. Notably, if we had used the single patch arrangement shown in Figure 3b, then the landscape characteristics along the edges of each patch would have been lost.

Therefore, we also produced a second patch arrangement, identical to the first but shifted by two kilometers to the east and two kilometers to the south. This operation returned the blue patches shown in Figure 3c. In this way, the total data volume is also increased, providing multiple terrain and landslide scenarios defined over the different spatial data structures.

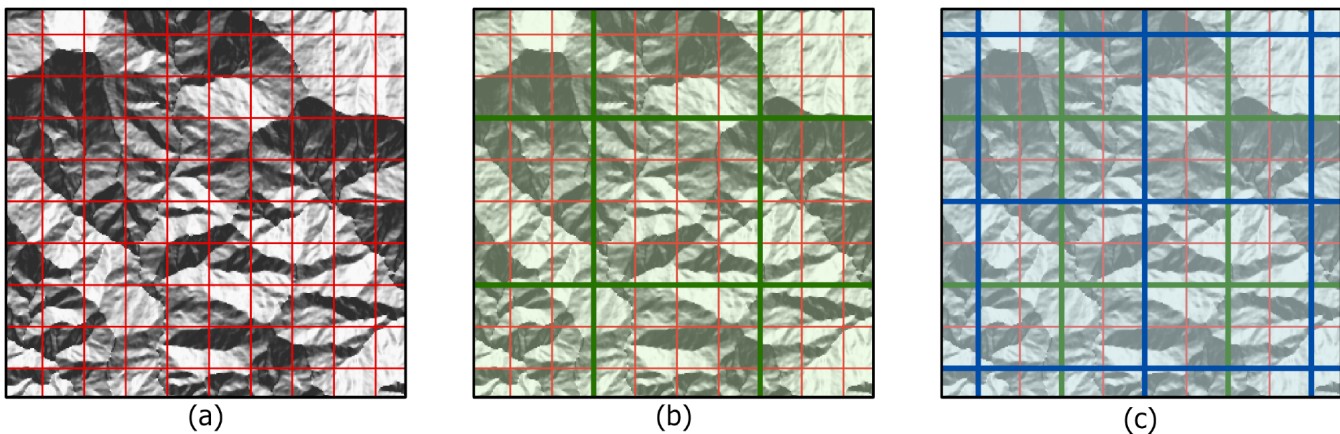

|  (a)  |  (b)  |  (c)  |

**Figure 3.** Panels showing the various mapping units structures: *(a)* the covariate and existing inventory grid structure, with 1 $\times$ 1 km. grid with $32 \times 32$ pixels of terrian image in the background *(b)* the patching of 4x4 inventory grid with $4 \times 4$ km. grid and *(c)* the shifted patch structure with similar grid structure as *(b)*.

Note that these spatial manipulation procedures are quite common for Convolutional Neural Networks (e.g., Amit and Aoki, 2017). Here, we have simply adapted them in the context of the gridded structure defined by Kincey et al. (2021).

## 5 Predictors

The predictor set we chose features a number of terrain attributes, as well as hydrological and seismic factors. These predictors are selected based on their influence on landslides, which is observed by many existing works as represented in the table 1. Our assumption is that their combined information is able to explain the distribution of landslide occurrences and area densities (the combined targets of our ENN) both in space and time. These predictors are listed in Table 1, graphically shown in Figure 4. Below, we report a brief explanation to justify their choice.

**Table 1.** Predictors' summary

| Type | Covariate:Acronym \| Unit | Reference |
|---|---|---|
| Morphometric (30 m SRTM) | *Slope (Slope \| degrees)* | (Zevenbergen and Thorne, 1987) |
| Morphometric (30 m SRTM) | *Elevation (Elevation \| meters)* | – |
| Morphometric (30 m SRTM) | *Northness (Northness \| unitless)* | (Steger et al., 2016) |
| Morphometric (30 m SRTM) | *Eastness (Eastness \| unitless)* | (Steger et al., 2016) |
| Morphometric (30 m SRTM) | *Profile Curvature (PRC \| $m^{-1}$)* | (Heerdegen and Beran, 1982) |
| Morphometric (30 m SRTM) | *Planar Curvature (PLC \| $m^{-1}$)* | (Heerdegen and Beran, 1982) |
| Morphometric (30 m SRTM) | *Topographic Wetness Index (TWI \| unitless)* | (Sörensen et al., 2006) |
| Precipitation (∼5km CHRIPS) | *Maximum daily rainfall (Max. Precip.\| mm/day)* | (Funk et al., 2015) |
| Precipitation (∼5km CHRIPS) | *95% CI rainfall in the inventory period (95% CI Precip. \| mm/day)* | (Funk et al., 2015) |
| Seismic shaking (1 km USGS) | *Maximum Peak Ground Acceleration from main event and major aftershock (Max PGA \| $m/s^2$)* | (Worden and Wald, 2016) |
| Seismic shaking (1 km USGS) | *St. Dev. Peak Ground Acceleration (1Std. PGA \| $m/s^2$)* | (Worden and Wald, 2016) |
| Distance to River | *Distance to River (Dist2Riv \| meters)* | – |
| Monsoons after Earthquake (count) | *Monsoons after the Earthquake (Monsoons \| year)* | – |

The *Slope* carries the signal of the gravitational pull acting on potentially unstable materials hanging along the topographic profile (Taylor, 1948). *Elevation*, *Eastness* and *Northness* are common proxies for a series of processes such as moisture, vegetation and temperature (Clinton, 2003) and their effect on slope stability (Neaupane and Piantanakulchai, 2006; Whiteley et al., 2019; Loche et al., 2022). As for the *Planar* and *Profile Curvatures*, these are known to control the convergence and divergence of overland flows (Ohlmacher, 2007). This hydrological information is also supported by *Topographic Wetness Index* and *Distance to River* (Yesilnacar and Topal, 2005). To these finely represented predictors, we also added a number of coarser ones, representing the potential triggers behind a landslide genetic process, namely, *Rainfall* (its Maximum value (per pixel) and 95% Confidence Interval (CI) (per pixel) within the inventory time-frame, calculated from daily CHIRPS data spanning between two subsequent landslide inventories; Funk et al., 2015) and *Peak Ground Acceleration* (its Maximum value from between mainshock and aftershock and their respective standard deviation estimated using empirical ground motion prediction equations, available through the ShakeMap system of the United States Geological Survey (USGS); Worden and Wald, 2016).

The PGA is empirically estimated from local ground motion recording stations, and it has been shown to correlate to the Gorkha coseismic landslide scenario (Dahal et al., 2023). Similar to PGA, Peak Ground Velocity (PGV) could also be used in this case to model the landslides as they are better predictors in some cases(Maufroy et al., 2015; von Specht et al., 2019). However, very few stations actively recorded the Gorkha earthquake. This is the reason why the differences in the spatial

patterns of the PGA and PGV available in the USGS ShakeMap system are negligible (Hough et al., 2016). Therefore, with the objective of explaining the coseismic landslide scenario over such a large study area, any of the two shakemaps would produce similar results.

To these spatially and temporally varying predictors, we also added the monsoons' count after the Gorkha Earthquake to inform the model about the combined effect of landscape characteristics, earthquake legacy and meteorological stress.

The code to prepare these datasets using the Google Earth Engine is available at link.

# 6 Neural networks

## 6.1 Model architecture

To contextually estimate landslide susceptibility and area density, we designed an NN with a multi-output design, relying on the same 1 km gridded data input. In short, the first model component estimates a "pseudo-probability" via a sigmoid function, whereas the second component regresses the area density information against the same set of predictors used in the previous step.

The NN design is shown in the Figure 5. The susceptibility block is modified from the U-Net model (Ronneberger et al., 2015) with the backbone of Resent18 (He et al., 2015), where the model processes input information through the 18 blocks of Convolution, Batch Normalization (Ioffe and Szegedy, 2015a), dropout, Rectified Linear Unit (ReLU) and Max pooling (Wu and Gu, 2015) with a total 23,556,931 number of trainable parameters which are variables that need to be optimized during the training process. The convolution layer in each block convolved with the $3 \times 3$ window, and it was initialized with the Glorot uniform initialization function (Glorot and Bengio, 2010). The convolution function was followed by a batch normalization which works as a regularization function. This prevents the model from overfitting, and it is followed by a dropout layer. The dropout layer randomly de-activates 30% of the neurones in the convolution layer, such that the model does not overfit. Following this, the feature space passes through the ReLU activation function, which allows for non-linearity in the model and finally, a max pooling layer is added to reduce the spatial dimension of the feature space.

The decoder part consists of the U-Net structure, but unlike the conventional U-Net model, it produces an output scaled down by a factor of 8. The schematic design of the model is shown in Figure 6. To understand the spatial dependence between the different inventory grids ($1 \times 1$ km$^2$ grid), we have used a $4 \times 4$ aggregation patch as input for the susceptibility block, which is equivalent to $128 \times 128$ input pixels. After receiving $128 \times 128$ pixels, the convolution operation learns the contribution of physical properties such as earthquake and rainfall intensities as well as terrain characteristics to produce the susceptibility in a $4 \times 4 \times 1$ batch of $1 \times 1$ km$^2$ grids. We stress here that we specifically chose to use a $32 \times 32$ pixel structure per 1 km grid to convey all the possible information to the model and provide flexibility to the neural network to learn relevant information. As a result, the model can extract the relevant information it needs from the distribution of $32 \times 32$ pixels, rather than using arbitrary summary statistics such as the mean and standard deviation as per tradition in the geoscientific literature (e.g., Guzzetti et al., 2000; Lombardo and Tanyas, 2020). In other words, the model can learn by itself: (1) scanning $32 \times 32$ pixel images corresponding to single 1 km grid cells and (2) matching the image characteristics to the landslide presence/absence labels.

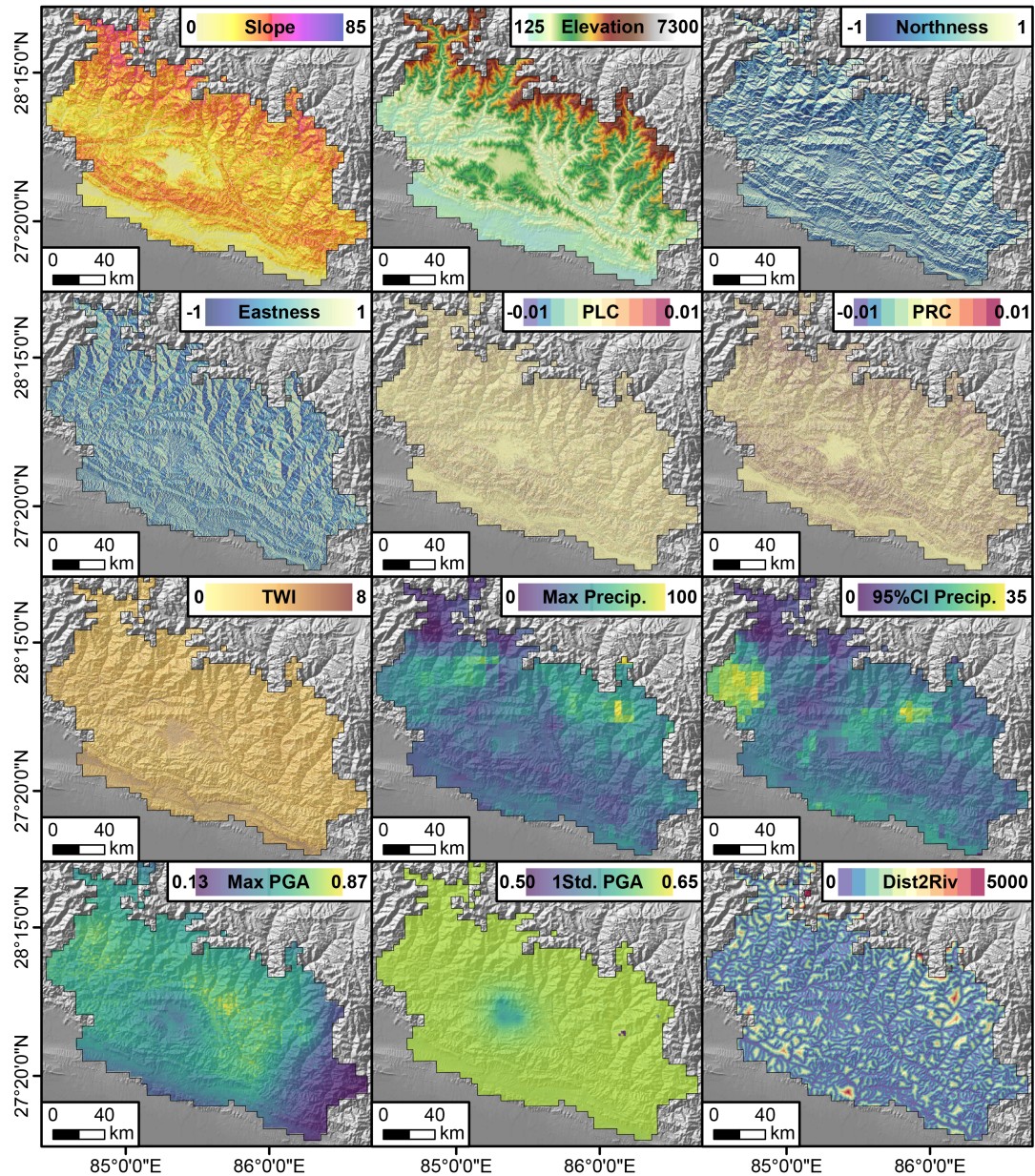

**Figure 4.** Predictors used for training the Ensemble Neural Network. The Max Precip. is one example of the maximum daily rain calculated for each inventory. The same applies to the 95% CI Precip. calculated as the difference between the 97.5 and 2.5 percentiles of the daily rainfall distribution. Max PGA and 1Std PGA are respectively the maximum and one standard deviation calculated from the Peak Ground Acceleration maps of the main and after shocks. Dist2Riv is the Euclidean distance from each 30m pixel to the nearest streamline. PLC, PRC and TWI are acronyms for Planar Curvature, Profile Curvature and Topographic Wetness Index.

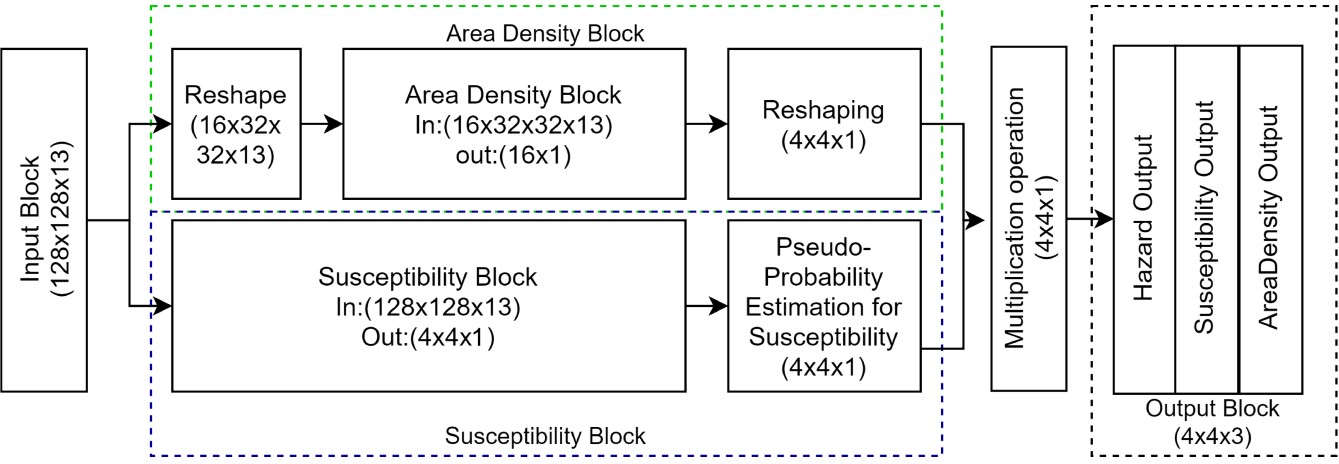

**Figure 5.** Designed landslide susceptibility and area density prediction model.

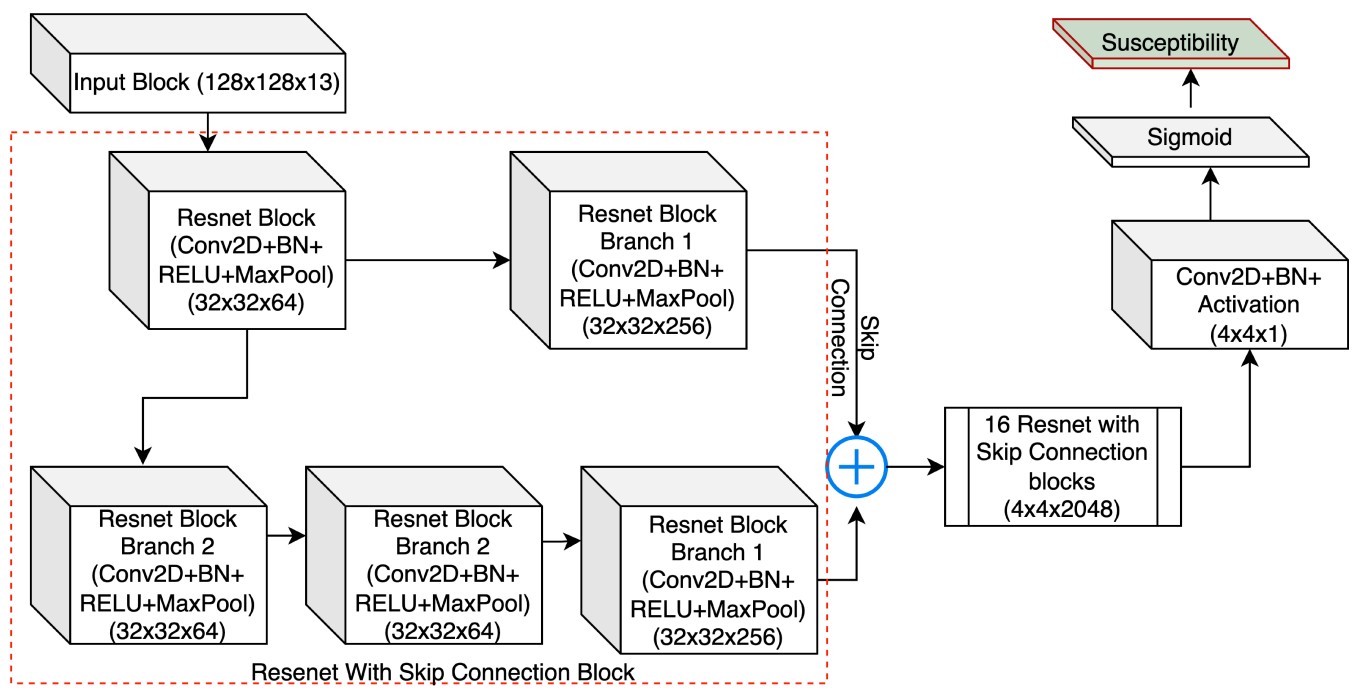

**Figure 6.** Susceptibility part of the model designed with U-Net like structure.

The area density block also relies on a 1 km grid structure. However, we did not introduce the $4 \times 4 \times 1$ neighborhood because the landslide presence/absence data presents some spatial pattern beyond the extent of a 1 km grid. Conversely, the landslide area data does not present obvious spatial clusters of small or large densities.

Furthermore, it is also evident that landslides are discrete phenomena in space. This means that a large area density can be estimated at a 1 km grid, but its neighbor may not have suffered from slope failures (area density = 0). Conveying this "salt and pepper" spatial structure to the U-Net (via a $4 \times 4$ neighboring window) tasked with regressing continuous data would negatively affect the model (unreported tests).

To address this issue, we reshaped the input data to a $16 \times 32 \times 32 \times 13$ shape, where 16 inventory grids, each associated with 13 predictors of $32 \times 32$ pixel size, are present. The area density block is made of six dense sub-blocks, encompassing fully connected, batch normalisation (Ioffe and Szegedy, 2015a) and dropout layers (Srivastava et al., 2014a). Before passing the data to the dense block, we added one Convolution block consisting of Convolution, Batch Normalization (Ioffe and Szegedy, 2015a) as well as Rectified Linear Unit and Max pooling (Wu and Gu, 2015) layers to extract the features from the input patches. Once both the area density and the susceptibility are estimated, the area density needs to be reshaped to match the data structure of the susceptibility component. To then generate landslide hazard estimates, as per the definition proposed by Guzzetti et al. (1999), we added a step where the pseudo-probability of landslide occurrence is multiplied by the landslide area density.

Notably, the developed model is spatio-temporal because it is built to explain the variability of the landslide hazard in space and time. However, the convolution layers used in this modelling approach are of a 2D nature. Therefore, the spatial structure in the data is handled via the 2D CNN, whereas the connection between subsequent spatial layers is not explicitly built-in. The landslide hazard definition includes the concept of return periods to treat the temporal component, where for a given triggering situation of a given return period, the landslide hazard is estimated. The concept of return time could theoretically included as part of the architecture presented here. However, the length of the multi-temporal inventory is relatively short. Thus, our model should not be considered suitable for long-term prediction, but rather, its validity is confined to the spatio-temporal domain under consideration or very close to it, both in space and time (Wang et al., 2023). In other words, it should not be considered for generating long-term predictions under climate change scenarios.

## 6.2 Experimental setup

Binary classifiers are quite standard in machine/deep learning. Thus, for the susceptibility component, we opted for a focal Traversky loss function ($FTL_c$, see equation below for clarity), as Abraham and Khan (2018) have shown this measure to be particularly suited for imbalanced binary datasets such as ours. The major reason to choose this loss function is the dominant absence of landslides in the dataset, complemented by much fewer cases ($\approx 10\%$) where slope failures occurred. This might bias the model result and lead to a wrongly trained model if the loss function is not suitably implemented to handle this imbalance. The definition of Focal Traversky Loss can be denoted as:

$$FTL_c = \sum_c (1 - TI_c)^{\frac{1}{\gamma}},$$

$$TI_c = \frac{\sum_{i=1}^{N} p_{ic} g_{ic} + \epsilon}{\sum_{i=1}^{N} p_{ic} g_{ic} + \alpha \sum_{i=1}^{N} p_{ic} g_{ic} + \beta \sum_{i=1}^{N} p_{ic} g_{i\bar{c}} + \epsilon},$$

(1)

where, $\gamma$ is focal parameter, $p_{ic}$ is the probability that the pixel $i$ is of the Landslide class $c$ and $p_{i\bar{c}}$ is the probability that the pixel $i$ is of the non-landslide class $\bar{c}$. The same holds for $g_{ic}$ and $g_{i\bar{c}}$. $\alpha$, and $\beta$ are the hyperparameters which can penalize false positives and false negatives and $\epsilon$ value was set to $1^{-7}$. Furthermore, $c$ represents the landslide presence class in the landslide classification problem; this could be represented by any positive integer in the case of multiclass classification problems. As for $FTL_c$, it represents the Focal Traversky Loss for binary classification, and $TI_c$ is the Traversky Index.

To train the susceptibility component of the model, we trained a standard U-Net equipped with an early stopping functionality for a total of 500 epochs. The stopping criterion was set to detect overfit that may last for over ten epochs. The overall data was then randomly split into training and testing sets to monitor the U-Net learning process.

As for the area density component, we opted for a loss function expressed in terms of mean absolute error ($MAE$, see equation below for clarity), following the recommendations in Qi et al. (2020). MAE is denoted as:

$$\text{MAE} = \frac{\sum_{i=1}^{n} |y_i - \hat{y_i}|}{n}$$

(2)

where, $y_i$ is the observed area density and the $\hat{y_i}$ is predicted area density in the $i-th$
pixel, and n is the total number of samples in one batch.

To train the area density component, the imbalance in zeros and ones hindered the optimisation process because the mean absolute error function did not perform well when most of the landslide densities were zeroes. This led to exploding gradients, returning all zero as the output. To solve this issue, we gradually increased the complexity of the task by subsampling the data and transforming the distribution of area density. The process is commonly known as curriculum learning (Wang et al., 2021), which lets the model learn a simple task at the start, and the process continues by gradually increasing the complexity of the subsequent tasks, each one linked to the previous one. To do so, we first removed all data points which contained zeros among the area density 1 km grids. We then log-transformed the target variable to convert the exponential-like distribution to a gaussian like distribution. Once the data was expressed according to a near-normal distribution, we trained the model for 200 epochs, including an early stopping criterion. The estimated parameters were set to initialize the subsequent steps. Specifically, with the initialisation parameters available, we removed the logarithmic transformation and trained the model directly in the original landslide area density scale. This step was further run over 200 epochs, and the resulting parameters were fine-tuned to match the overall landslide area density distribution. In other words, we re-introduced the 1 km grids with zero density at this stage. Ultimately, the data was then randomly divided into 70% for calibration and 30% for validation.

The models were trained in a machine with 160 GB Random Access Memory, a 32-core AMD Ryzen Threadripper PRO virtual CPU, and an NVIDIA RTX A4000 GPU. The computational resource used in this case made use of a shared infrastructure, with the entire training process taking 30-40% load on the node and taking 35-40 hours, depending on the available

GPU memory. For the backpropagation, we used the Adam optimiser (Kingma and Ba, 2014), with an initial learning rate of $1e^{-3}$, exponentially decreasing every 1000 steps of training. Because simultaneously training a model with two outputs based on a large and complex dataset would be extremely difficult to achieve, we opted to train the two elements separately in the beginning and combine their weights at the end of the learning process to generate a single model. This is then further trained for a few more steps to optimize the area density component for the non-landslide grids. This approach is commonly known as ensemble modelling in the data-driven modelling context. The model is trained with a batch size of 16, and the training dataset is randomly divided into a 30% validation set to check the model convergence at each epoch. The training process also featured an early stopping functionality where the model training would stop if the validation loss started to diverge. Simply put, the model weights were selected at the minimum validation loss to avoid overfitting. The number of convolution blocks, batch size, and initial learning rate were optimised through a hyperparameter tuning process, whilst other parameters were selected from the pre-existing models. For the convolution blocks, all the integer numbers of blocks were tested between 6 and 64, and we found that 18 was the best-suited number of blocks. For the batch size, four different batch sizes were tested (8, 16, 32, 64), and the fastest convergence was obtained through a batch size of 16. Moreover, learning rates from 1 to $1e^{-4}$ were tested by decreasing the learning rate by a factor of 10 each time, and we found the initial learning rate of $1e^{-3}$ to be the most stable.

## 6.3 Performance metrics

We used the following performance metrics for susceptibility and the area density components.

### 6.3.1 Susceptibility component

To evaluate the model's performance during the training process and the inference, we used two common metrics, namely the F1 score (Meena et al., 2022; Nava et al., 2022) and the Intersection over Union (IOU) score (e.g., Huang et al., 2019; Ghorbanzadeh et al., 2020). We did not use binary accuracy because it is heavily influenced by data imbalance (Yeon et al., 2010; Li et al., 2022) and can produce high accuracy, even for poor classifications. The F1 score (3) is calculated as:

$$
F1 = \frac{2 \times \text{precision} \times \text{recall}}{\text{precision} + \text{recall}},
$$
$$
\text{precision} = \frac{TP}{TP+FP}, \text{ recall} = \frac{TP}{TP+FN}, \tag{3}
$$

where, TP denotes the True Positive, FP denotes the False Positive, TN denotes True Negative and FN denotes the False Negative in the confusion matrix.

As for the IOU, this is another common metric for binary classifiers, computed as:

$$
IOU = \frac{TP}{TP+FN+FP}, \tag{4}
$$

where, TP denotes the True Positive, FP denotes the False Positive, TN denotes True Negative and FN denotes the False Negative in the confusion matrix.

We chose to use the IOU because it is a metric specifically dedicated to highlighting the accuracy in predicting the number of susceptible pixels and their location in a raster image (Monaco et al., 2020). Furthermore, to visualize how the model performs at different probability thresholds and what the performance capacity of the model is we also evaluated the Receiver Operating Characteristic (ROC, Fawcett, 2006) curve. This is generated at varying probability thresholds by computing pairs of True Positive and False Positive Rates. Moreover, we calculated the Area Under the ROC Curve (AUC) to evaluate the model's performance and to observe if the model overfits (Brenning, 2008; Brock et al., 2020).

### 6.3.2 Area density component

To evaluate the training process for the landslide area density, we opted to use the MAE (see Eq. 5) to monitor how the algorithm converges to its best solution, minimising such parameter. During the inference process, we also considered the Pearson's R coefficient Pearson (1895), defined as:

$$R = \frac{\sum (x_i - \bar{x})(y_i - \bar{y})}{\sqrt{\sum (x_i - \bar{x})^2 \sum (y_i - \bar{y})^2}},$$
(5)

where, $R$ = correlation coefficient, $x_i$ = values of the $x$-variable in a sample, $\bar{x}$ = mean of the values of the $x$-variable, $y_i$ = values of the $y$-variable in a sample, $\bar{y}$ = mean of the values of the $y$-variable.

This parameter essentially provides the degree of correlation between two datasets, i.e., the observed and predicted landslide density per 1 km grid. A perfect model should have Pearson's R-value of 1, whereas two totally uncorrelated vectors would return a Pearson's R-value of 0.

## 7 Results

This section reports the model performance, initially from a purely numerical perspective. Later, we will translate this information back into maps and evaluate their temporal characteristics.

Figure 7 offers an overview of the performance our ENN returned for its two components. The left panel reports an AUC of 0.93, associated with an F1 Score of 0.96 and IOU of 0.95. This predictive performance complies with the classification performance of outstanding data-driven models (Hosmer and Lemeshow, 2000). In the context of NNs, this is quite common because such architectures as much as other machine/deep learning tools and advanced statistical methods have proven to be able to reliably classify a landscape into landslide prone/unstable slopes (e.g., Lombardo et al., 2019; Steger et al., 2021). Traditionally, the only missing element is that the vast majority of efforts so far have been spent solely in the context of pure spatial predictions, whereas the temporal dimension has been explored in a relatively smaller number of multivariate applications (Samia et al., 2017; Fang et al., 2023). Conversely, the performance of the area density component are far beyond the few analogous examples in the literature. So far, no spatially nor temporally explicit model exists for landslide area density. However, four recent articles have explored the capacity of predicting landslide areas (Lombardo et al., 2021; Aguilera et al., 2022; Bryce et al., 2022; Zapata et al., 2023). They all returned suitable predictive performance, but still far from the match

seen in the second panel of Figure 7, between observed and predicted landslide density. There, an outstanding alignment along the 45 degree line is clearly visible, together with a Pearson's R coefficient of 0.93 and a MAE of 0.26%. It is important to stress that such metrics are calculated including the 1 km grids with zero landslide densities, i.e., the validation set in the study area as a whole. We also computed the same metrics exclusively at grid cells with a positive density, these resulting in a Pearson's R coefficient of 0.92 and a MAE of 0.24%.

At a closer look, we can note a few exceptions, with some observations being strongly underestimated and very few cases being overestimated. This might be because we used MAE as the loss function. The MAE is a metric tailored towards the mean of a distribution, and therefore smaller values in the batch may be misrepresented without increasing the MAE. Moreover, the model is optimized by minimization the MAE. Thus, it places more emphasis on suitably estimating large landslides and potentially underestimation the smaller ones. This problem could also have been influenced by the log-transformation step introduced at the beginning of the training process. This inevitably converted smaller values to very small ones, thus limiting their influence on the loss function even more. We are sharing this issue with the reader to provide the best description of a new modeling protocol. However, we should also mention that we consider such misrepresentation a minor problem. In fact, geoscience and risk science in general, refer to the worst-case scenarios as the prediction target. Here this is expressed by very large landslides, which appear to be correctly represented in most cases. Another worst-case scenario may be the combination of large numbers of medium-sized landslides and their potential coalescing evolution. However, the bulk of the landslide density distribution is very well represented. This leaves most of the errors confined to the left tail (or very small) of the landslide density distribution. These correspond to the phenomena from which one would expect the least potential threat or capacity to create damage, assuming a uniform vulnerability distribution.

These two plots offer a graphical overview of our ENN performance but they do not convey their signal in space and time. To offer a geographic and temporal overview of the same information, we opted to translate the match between observed and predicted values into maps, both for the susceptibility and area density components. Figure 8 shows confusion maps (Titti et al., 2022; Prakash et al., 2021), where the distribution of TP, TN, FP, FN is geographically presented for the coseismic susceptibility as well as the following seven post-seismic scenarios. Across the whole sequence of maps, what stands out the most is that the TP and TN largely dominate the landscape, with few local exceptions. Notably, aside from the geographic translation of the confusion matrix, we reported the actual counts in logarithmic scale through the nested subpanels. There, the dominant number of TP and TN is confirmed once more and a better insight is provided on the model misses (FP and FN).

Figure 9 highlights the mismatch between observed and predicted landslide area densities. Most of the residuals are confined between -1 and +1 percent, with a negligible number of exceptions reaching an overestimation of -45% and an underestimation of +15%. Aside from these outliers, the most interesting element that stands out among these maps is the fact that the residuals do not exhibit any spatial pattern. They actually appear to be distributed randomly both in space and time.

Having stressed the predictive performance reached by our ENN, in Figure 10, we finally offer a direct overview of the two outcomes (susceptibility and area density) as well as their product (hazard). Figure 10 reports the co-seismic case only and the post-monsoon estimates. We opted for this for reasons of practicality and visibility in a quite crowded subpaneled figure. To accommodate for the potential curiosity of the readers, we recall here that code and data are open and accessible at this link.

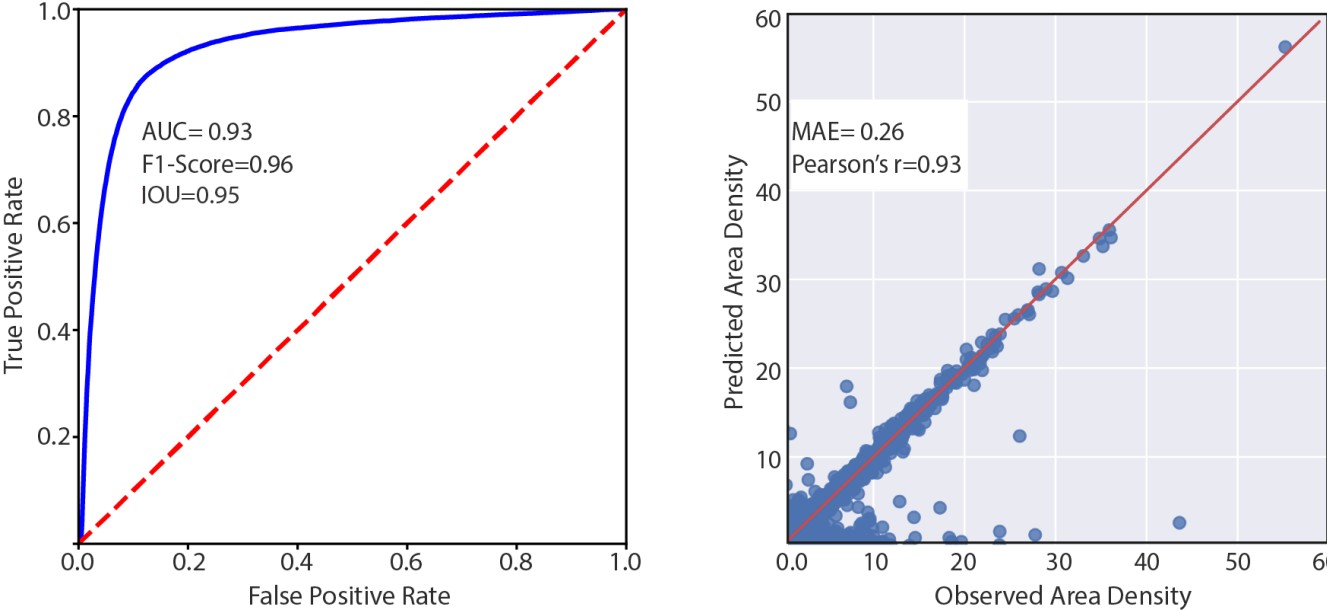

**Figure 7.** Summary of model's performance for the two components: landslide susceptibility in the left panel and Area Density in the right panel in the validation data.

Reading these maps should be intuitive, but below we stress the assumptions behind the hazard one, being the first time such a map has ever been shown. The first column reports the probabilities of landslide occurrence per 1 km grid. The second column shows the predicted landslide area density for the same 1 km lattice. The product of the two delivers an important element, where only coinciding high susceptibility and high density grids stand out. The rationale behind this is that large probability values of landslide occurrence will be inevitably canceled out whenever multiplied by low area density values. The same is valid in the opposite case. Large expected densities will be canceled out if multiplied by very low susceptibility values. Thus, the hazard maps really do inform of the level of threat one may incur at certain 1 km grids and certain times, because a high hazard value implies that the mapping unit under consideration is not only expected to be unstable but the resulting instability is expected to lead to a large failure, too.

The implications of the estimated patterns and considerations in terms of hazard will be further explored in Section 8. To support such discussions and highlight the link between susceptibility, hazard and their temporal evolution, we opted to plot their signal via two-dimensional density plots, these being shown in Figure 11.

We can observe an interesting element, attributable to a concept known as earthquake legacy in the geoscientific literature. In fact, high landslide area density values associated with high susceptibility are quite represented on the coseismic panel and the first post-seismic one. However, as time passes, the density and proneness of the landscape appears to be estimated with lower landslide susceptibilities and densities.

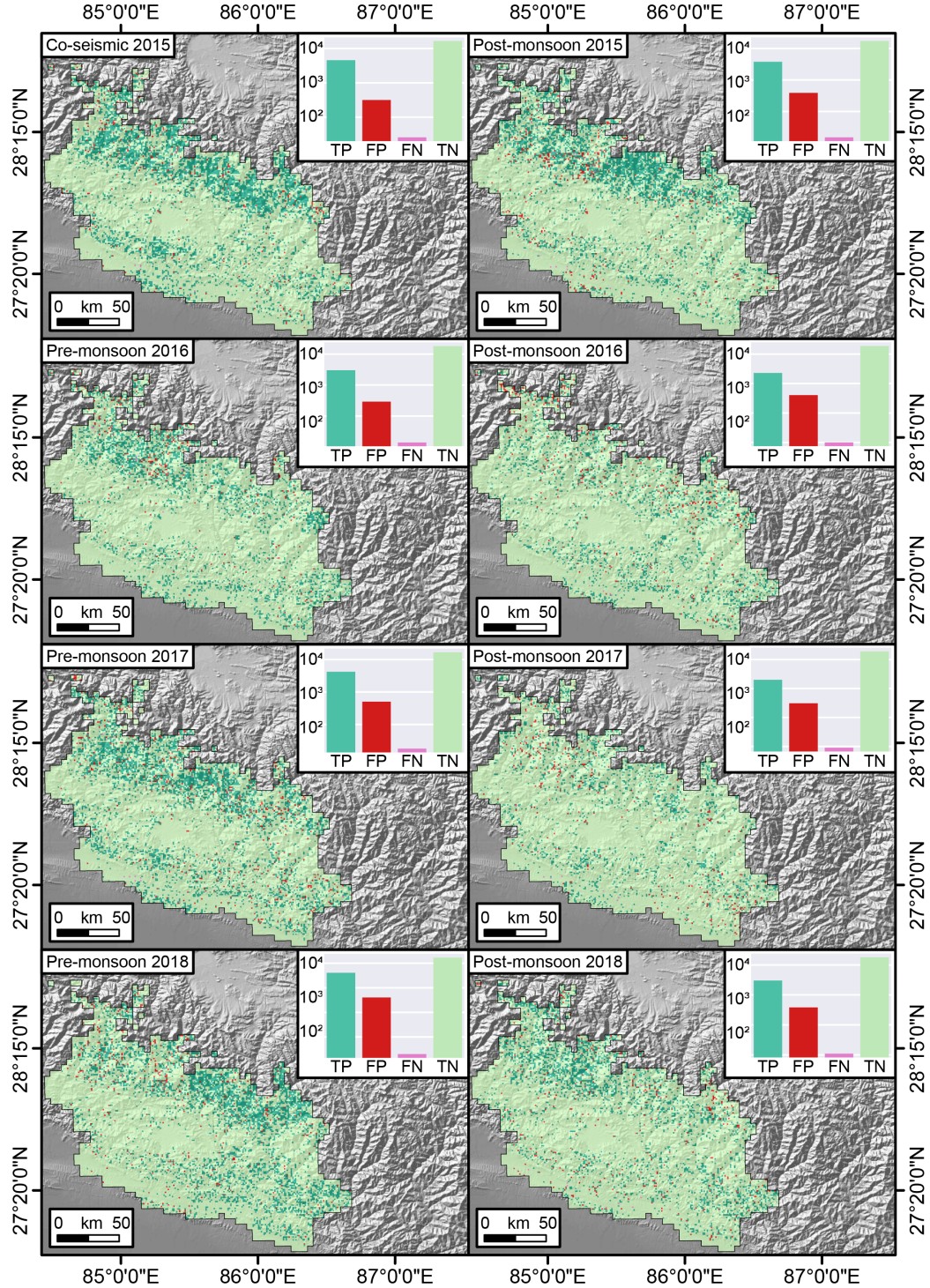

**Figure 8.** Confusion Maps offering a cartographic predictive of the performance for the susceptibility component. the TP, FP, FN and TN are represented in the log scale.

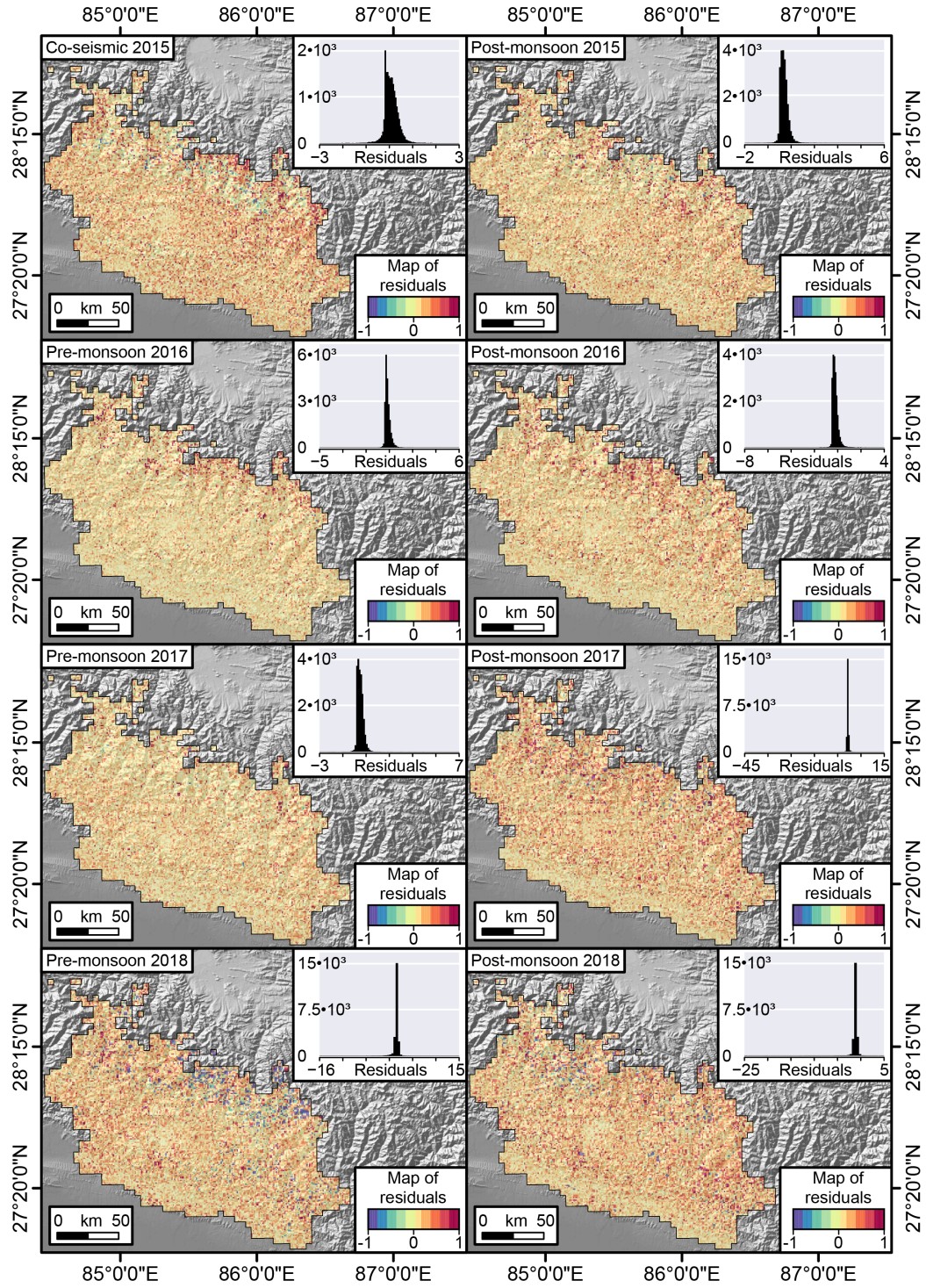

**Figure 9.** Maps displaying the pre- and post- monsoon residuals for the area density (expressed as percentages). The residuals are computed as observed landslide density minus the corresponding predicted values.

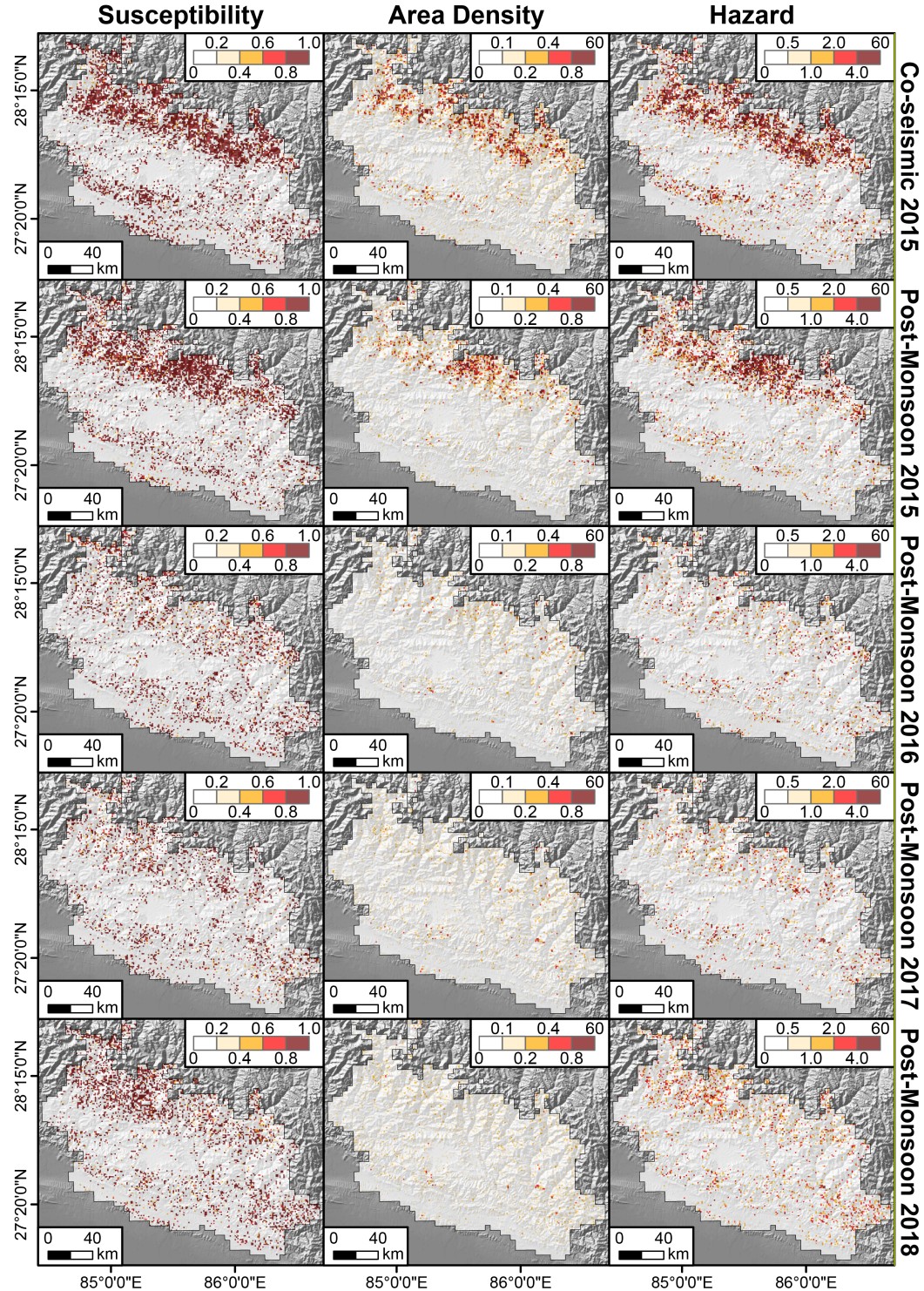

**Figure 10.** Predicted landslide susceptibility, area density and hazard over time for Post Monsoon seasons only, because those period had most of the landslide.

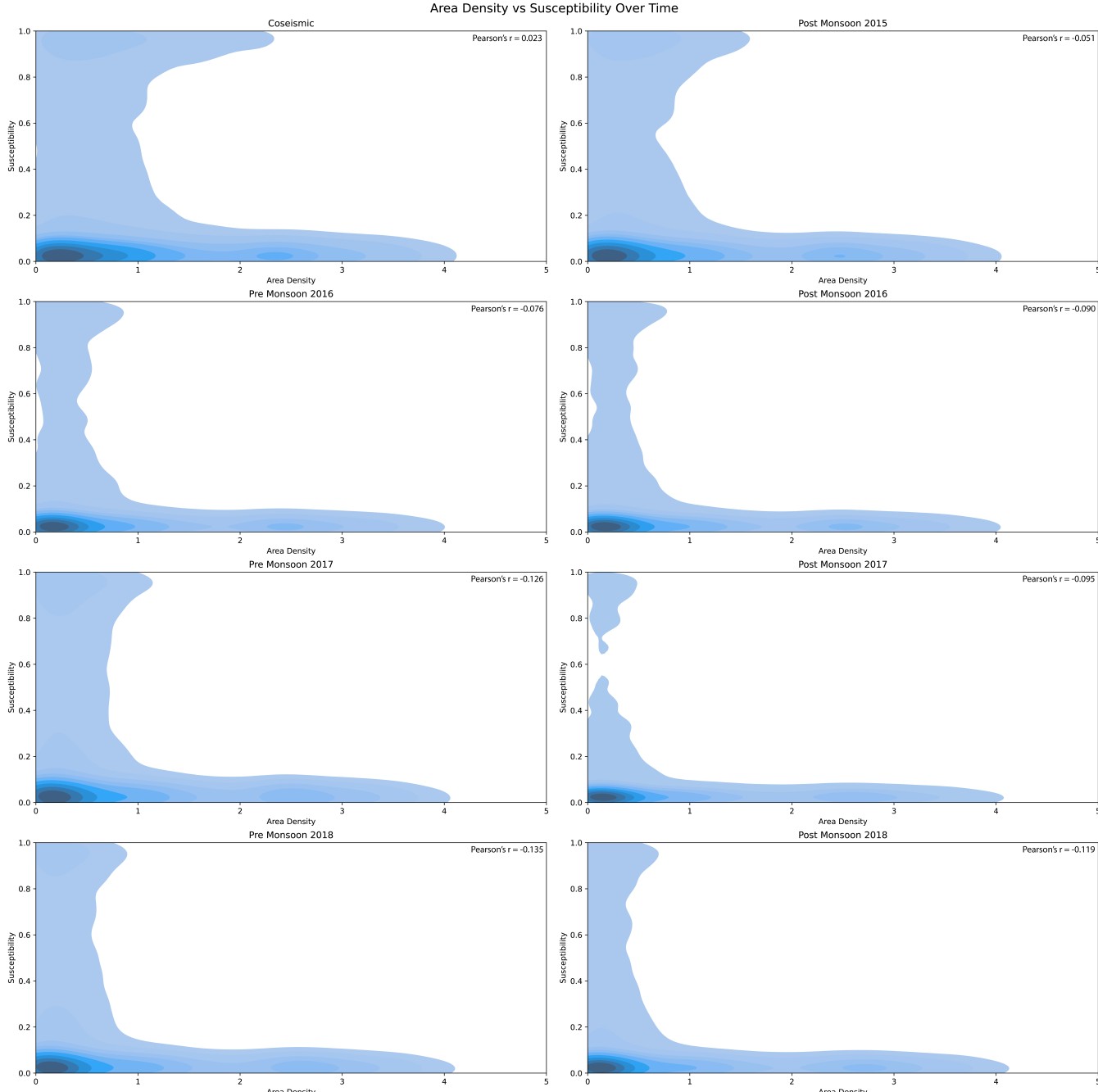

**Figure 11.** Contour plot of Area Density versus Susceptibility in different time periods showing how the area density and susceptibility are related to each other. Where, lighter color represents the lower desnity of the values and darker color represents the higher density of the values. Furthermore, it shows how in different period after the earthquake the area density and susceptibility are distributed over space and how the range of susceptibility and area density changes.

## 8 Discussion

In this section, we discuss the model's performance, applicability, limitations and necessary future developments in two sub-sections containing the supporting and opposing arguments.

### 8.1 Supporting arguments

The model results and the observations show that the deep learning-based methods perform well in predicting landslide susceptibility and area density through a joint modelling approach. Such models can obviously provide much more information than modelling only susceptibility (Lombardo et al., 2021). Only using the susceptibility information is blind to landslide characteristics, such as how many landslides may manifest or how large they may become once they start moving downhill (Di Napoli et al., 2023). Thus, the combined information of which slope may be considered unstable and the expectation on the landslide can become an important source of information not only for hazard assessment but even for risk reduction and management practitioners, once combined with the elements at risk.

Our ENN has shown the capacity to assess the two core elements, and interesting considerations can be made about its outcome. Figure 8 shows that each inventory mostly produced True Positives and Negatives across the whole study site. More importantly, the number of false negatives was almost negligible. As for the False Positives, their number is reasonable and highlights locations where landslides have not manifested yet but may still occur in the future. As for the area component, Figure 9 shows that the patterns of the residuals appear quite random both in space and time, thus fulfilling the ideal homoscedasticity requirements of a data-driven model. We can also stress that most of the residuals away from a few percentage points are confined towards negative values. This implies that our model overestimates the landslide area in a few isolated cases. However, similarly to the point raised for the FP in Figure 8, this outcome is to be expected. A negative residual indicates a location where the observed landslide area is lower than the predicted one. As most of the study site is characterized by grid cells where landslides did not occur, a negative residual points out at locations that may not have exhibited landslides in the first place but whose geomorphological characteristics still indicate a likely release of a relatively larger unstable mass in the future.

Ultimately, Figure 10 shows the constructive and destructive interference between the susceptibility and area density signals. This leads to isolating landslide hazardous locations, which appear to be mostly located along the highest portions of the Himalayan range under consideration. There, a greater hazard is to be reasonably expected, for the higher relief is associated with higher gravitational potential and thus with a greater conversion into kinetic energy as the given landslide triggers, propagates, and finally halts.

An interesting by-product of our ENN can also be seen in Figure 11. There, the high hazard levels estimated for the first two landslide inventories are shown to decay with time. Moreover, we can also observe that the susceptibility and area density are not necessarily correlated, meaning that the probability of landslide occurrence is not directly correlated to its size. This was also visible in the raw data shared by Kincey et al. (2021). Such a decay supports the notion of earthquake legacy effects on landslide genetic processes (Ozturk, 2022), something still under debate in the geoscientific literature (Tanyaş et al., 2021b).

Our output could bring additional information on this topic, supporting the scientific debate on landslide recovery (the time required for a given landscape to go back to pre-earthquake susceptibility conditions) by observing the predicted susceptibility change over time. Overall, multi-temporal landslide inventories and various associated parameters (e.g., number, size, area or volume of landslides) have already been used to explore landslide recovery in post-seismic periods (eg., Tanyas et al., 2021). However, this has usually been done at a very generic and broad scale, leaving the slope scale usually out of the analytical process. Therefore, we see an added value in our model as it provides a comprehensive evaluation of landslide occurrences and their size. It is worth noting that examining the landslide recovery is beyond the scope of this research. Yet, something worth sharing with the readers is that the decay we observe appears to have slowed down in 2017 and 2018, with a slight increase in the number of landslides, susceptibility, area density, and hazard. During those years, though, Kincey et al. (2021) could not regularly map landslides as they previously did. Thus, both pre-monsoon seasons in 2017 and 2018 were mapped on a longer time window than the authors did in previous years, slightly inducing a temporal bias in the model.

Another element worth noting is that landslides across any given landscape are rare events. Thus, the number of presences will always be much smaller than the absences. This creates imbalanced data sets, which are often not ideally modeled in the deep learning context (see, Johnson and Khoshgoftaar, 2019). In turn, imbalanced data sets limit the use of traditional metrics such as accuracy and the use of loss functions such as Binary Cross Entropy, because the latter will produce high number of false negatives. We addressed this problem by adopting a Focal Taversky loss for the susceptibility component (Abraham and Khan, 2018). As for the area density component, we also faced some technical issues. Overall, around 85% of the 1 km grids cells had a zero density value assigned to them (no landslides). In addition to this issue, the area density distribution is quite positively skewed, and regression tasks in deep learning have been mostly tested in the context of Gaussian or near-Gaussian distributions. To solve this problem, we had to split the modeling routine into a series of intermediate operations. First, we removed all zeros and used logarithmic transformation to shape the data according to a normal distribution. From this, we trained the first stage of our area density component. Once the model converged to its best solution in the log-density domain, we interrupted the training procedure, removed the log transformation and further trained our model. This approach bypassed the need to implement even more complex NN architectures able to handle heavy-tailed distributions typical of extreme value theory (Weng et al., 2018).

An important factor to consider in a deep learning-based modelling approach is also the model overfitting and its generalization. Usually, large models can easily overfit and to avoid this, we have employed three major approaches. First, we used model regularization by adding batch normalization and dropout layers, according to standard deep learning practices (Srivastava et al., 2014b; Ioffe and Szegedy, 2015b). Secondly, we included an early stopping to halt the training process when divergence is observed in 30% random validation data. Up to this point, generalization has not been checked yet. This is actually achieved by further testing over 50% of the unseen data from which all the reported metrics are computed.

## 8.2 Opposing arguments

A major limitation we would like to recall relates to geological data availability. As discussed in the section 3, no detailed geological map covers the area where (Kincey et al., 2021) mapped landslides, nor does the only available alternative provide

enough geological classes to be meaningful in a data-driven context. In fact, the few available geological classes in Dahal (2012) present analogous proportions of landslide presence/absences and density data. This is mainly because the few geological units necessarily cover very large extents due to the coarse mapping resolution. In turn, this makes the available geology maps noninformative for the presented landslide modeling protocol. However, it is important to mention that a geological, or even better, a lithotechnical map constitute a precious layer of information for any engineering geological applications as they

control landslide initiation and size. This is the reason why we consider our space-time ENN as it is currently valid only for the area we tested it for. And we would recommend re-adapting it in case of new targets. For instance, another source of potentially relevant information could be the distance to active faults, which may be responsible for fracturing, fissuring and rock strength degradation in general. This being said, our model still produced outstanding performance (Hosmer and Lemeshow, 2000) within the context of the Nepalese landscape under consideration, although its validity elsewhere still needs to be verified.

Another limiting factor we faced was the selected multi-temporal landslide inventory. First, a sequence of three years after the earthquake may not be enough to display the potential of space-time landslide hazard models fully. Secondly, due to the resolution of satellite imagery used for mapping, some small landslides could have been omitted. To address the first issue, we have limited our predictor covariates to the temporal extent for which the mapping was carried out. This largely limits our model predictions to the time when ground truth data is available and verifiable. This model can capture the spatial and

temporal variations in the landslide occurrence and size and, therefore, can be used to understand the landslide hazard for different return periods. However, we have not calculated the landslide hazard for different return periods specifically because we did not have a sufficiently long landslide occurrence sequence. This could be further improved in subsequent studies as this work mainly focuses on introducing our model as a novel methodological tool. The second limitation related to the potential omission of smaller landslides is much more difficult to address mainly because, as universal functional approximators, the

deep learning models can only learn based on ground truth data. Therefore, this limitation cannot be removed. However, we recall that our model looks into a $1 \times 1$ km grid for the area density and presence and absence of the landslides. Therefore, the presence/absence of a landslide is not affected if a smaller landslide occurs together with larger and visible landslides, and it becomes a problem only when a small landslide occurs without a visible landslide within a $1 \times 1$ km grid. In other words, the mapping units of the susceptibility component should be assigned with a presence value, even if in a 1 km$^2$ grid, a few small

landslides may be missed. As for the area density, the effect on landslide area density may be more pronounced. However, as the landslide one may miss is small to very small in size, the effect may still be expressed within the uncertainty range of our model, potentially leading to minor omissions. This being said, outside the scope of this manuscript where this ENN architecture is introduced, we recommend future users to select an even more complete and long-term multitemporal inventory.

    From the pure methodological perspective, even though the model produced outstanding results, there is still much room

for improvement. As mentioned before, we addressed the heavy-tailed density distribution by using a log-transformation and L1 losses to measure the model convergence. In other words, we used a negative log-likelihood of a normal distribution to build our model, which in turn inherently assumes a normal distribution of the error. However, due to the fact that the area density follows an extreme value distribution in its right tail, instead of a model built on a log-transformation and then re-trained on the original density scale, a more straightforward procedure would directly use the original data distribution and

make use of performance metrics or losses that are suitable for the considered data. However, due to a lack of mature research on existing methods for using extreme value theory with deep learning, we could not use such an approach. For the further research, for instance, one of the possible approaches could be the integration of extreme value distributions (Davison and Huser, 2015) within our regression model. A similar procedure has been recently proposed to model wildfires (Richards et al., 2022; Cisneros et al., 2023).

Moreover, our model relies on a gridded partition of the geographic space under consideration. This lattice has two main elements that call for further improvements. The first is related to the size of the lattice itself. A 1 km grid cell is quite far from the spatial partition required to support landslide-risk-reduction actions. Thus, the current model output can offer a far richer information compared to the sole occurrence probabilities. However, to be actually useful for territorial management practices, the scale at which we trained should be probably downscaled at a finer resolution. The second element where our

ENN can be further improved in terms of spatial structure has to do with the geomorphological significance of a lattice when used to model landslides. Such geomorphological processes in fact, do not follow a regular gridded structure. In other words, when geoscientific go to the field, they do not see grids, whether they are few centimeter or the 1 km scale of our model. What a geomorphologist sees is a landscape partitioned into slopes. Slopes are also the same unit geotechnical solutions aim to address. Thus, an improvement to our ENN could involve moving away from a gridded spatial partition and towards more

geomorphological-oriented mapping units such as slope units (Alvioli et al., 2016; Tanyaş et al., 2022b), sub-catchments or catchments (Shou and Lin, 2020; Wang et al., 2022). Moreover, the modelling approaches could also be further improved by the addition of landslide trigger classification (see. Rana et al., 2021) which could inform the model about which parameter (either PGA or rainfall) is responsible for causing landslides in this particular predictive context.

It is important to stress here that the structure of a Convolutional Neural Network mostly requires gridded input data. Thus,

the extension towards irregular polygonal partitions such as the ones mentioned above would also require an adaptation of our ENN towards graph-based architectures (Scarselli et al., 2008).

Aside from the technical improvements we already envision, a key problem we could not address is the lack of detailed spatio-temporal information on roadworks. Landscapes, where roads are built, may relapse through pronounced mass wasting (Tanyaş et al., 2022a). Nepal is known for building small roads without accounting for the required engineering solutions

to maintain slope stability (McAdoo et al., 2018). For instance, Rosser et al. (2021) highlights that the elevated landslide susceptibility captured in post-seismic periods of the Gorkha earthquake could be partly associated with road construction projects. Thus, landslides trigger on steep slopes due to human interference, which we could not include in our model. During the very first phase of our model design, we actually tried to map those roads using freely available satellite images such as Sentinel 2 and PlanetScope. However, because the spatial resolution of those satellites is relatively coarse and the typical "self-

made" roads are quite small (2-3 meters in width), we could not automatize the road-mapping procedure to match our ENN spatio-temporal requirements. Therefore, rather than conveying wrong information to the model, we opted not to introduce road-network data to begin with. This is certainly a point to be improved in the future, not only for the Nepalese landscape but for any mountainous terrain where anthropogenic influence may bias the spatio-temporal distribution of landslides. Another parameter which could be added to inform our model about anthropogenic disturbances could be informal human modifications,

which have an influence on landslide trigger (Ozturk et al., 2022). Particularly, in this case, mountainous regions of Nepal do not have significant urban influences, and we opted not to include them. However, we recommend readers to include such features whenever necessary depending on the study sites.

We stress that the vast majority of Neural Networks are tailored towards solving prediction tasks, and our ENN essentially offered the same outstanding performances reported in many other deep learning applications. However, this architecture makes it very difficult to understand the causality behind the examined physical process. As our goal is to move towards a unified spatio-temporal hazard model, causality may not be a fundamental requirement at this stage. However, we envision future efforts to be directed towards more interpretable and causal machine/deep learning.

Ultimately, more can be done to clarify how our ENN should and should not be used, at least in its present form. For instance, with the current or even higher temporal frequency, our output could be used as part of parametric insurances (Horton, 2018) or large-scale risk reduction planning (Prabhakar et al., 2009). However, it is surely unsuitable for infrastructure planning (Dhital, 2000) for the 1 km$^2$ resolution is far too coarse to be useful for detailed scale design.

## 9   Conclusions

We present a data-driven model capable of estimating where and when landslides may occur, as well as the expected landslide area density per mapping unit, a proxy for intensity. We achieved such a modeling task thanks to an Ensemble Neural Network architecture, a structure that has not yet found its expression within the geoscientific literature, making this model the first of its kind. The implications of such a model can be groundbreaking because no data-driven model has provided an analogous level of information so far. The predictive ability of the model we propose still needs to be explored, isolating certain types of landslides, tectonic, climatic and geomorphological settings. If similar performance is confirmed, then this can even open up a completely different toolbox for decision-makers to work with. So far, territorial management institutions rely almost exclusively on susceptibility maps in the case of large regions and for long-term planning. The dependency on the concept of landslide susceptibility is also valid for regional and global organizations providing near-real-time or early warning alerts for seismically or climatically triggered landslides. The model we propose can potentially link these two elements and provide a piece of even richer information, exploiting its predictive power away from the six-month time resolution we tested here and more towards near-real-time or daily responses for various scale applications.

We conclude by stressing once more that we share data and codes in a GitHub repository accessible at this link to promote reproducibility and repeatability of the analyses presented in this work.

*Code and data availability.*   The data and code for the research can be accessed via https://github.com/ashokdahal/LandslideHazard

*Author contributions.* AD has implemented the model and run the whole analytical protocol. AD and LL have devised the research, produced scientific illustrations and written the manuscript. HT has helped through the pre-processing phase and contributed to writing the manuscript.

CvW, MvdM, PMM and RH has commented on the manuscript to improve it before submission.

*Competing interests.* There are no conflict of interest of any authors with the research or study area.

*Acknowledgements.* This article was supported by King Abdullah University of Science and Technology (KAUST) in Thuwal, Saudi Arabia, Grant URF/1/4338-01-01.

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
