# Peer review of "Space-time landslide hazard modeling via Ensemble Neural Networks"

_EGUsphere, 2023_

## Author Response (AR1)

**Editor Comments**

Dear authors

after revising the reviewers' comments and the discussion on the relevant portions of your manuscript, I believe that I can say that the work you have presented is innovative and of sure interest to the journal, but that it cannot be published in its present form. Although one of the reviewers (#1) declares himself quite satisfied with the submitted text, the comments raised by the second referee are relevant and cannot be discarded and underestimated. I therefore ask for your willingness to make changes with specific reference to the remarks of reviewer number 2. In particular, some questions are truly not of simple solution and raise issues that might seriously impact the validity of a very promising approach. Among them, I highlight the following.
* * *
Dear editor, thank you for your positive feedback and time spent evaluating our manuscript. We fully understand your concern and suggestions. In this version of the manuscript we have made changes according to the suggestions made by you, specifically addressing the reviewer's concerns. We believe that following the revision process, we have significantly improved the manuscript and for that, we would like to genuinely thank you and the reviewer.
* * *
1. The question of why geology (and, consequently, mechanical properties) are not considered relevant is truly something that deserves more explanation. It is true that very often we are faced with poor-resolution maps (especially in developing countries), but this does not allow us to bypass the problem or neglect the impact of considering terrain as having the same physical properties. For example, is there any way to try to quantify the errors induced by the uncertainty in the available geological maps? Is there any way to incorporate geologically-constrained variables that may be used as proxies for mechanical properties? Theoretically, (but also practically) any limit equilibrium model deeply relies upon geologically bounded properties and, all other conditions being constant, only soil/rock strength may explain slope failure. Could you please extend this into your analysis at least in the discussion?
* * *
Thank you for this insightful comment. We fully agree with your comment that geology is a crucial factor in slope stability assessments. In this case, however, the geological map present in the Nepalese territory from Dahal et al. (2006) largely excludes upper Himalayan territory (i.e., no-data) and the other geological map (provided by USGS) is much more generic, to the point of having only four classes. As it is evident, the Himalayan Geology is much more complex than this, and host many more than just four lithotypes. Therefore, these maps could not fully represent the complexity of the geological outcrops.

Therefore, as partial as it may be, we made use here of the geological map produced by Dahal et al. (2006). Again, this is not covering the entirety of the study area but we will use it to highlight a characteristic of the map that would in some way support its exclusion from the model. As you can see, below we present the landslide size distribution per geological class. Essentially, none of these classes behaves significantly differently from the other. The only one that shows a significant difference in landslide area (and here significant has a traditional statistical connotation) is Phulchowki formation. Every other boxplot is mostly contained in the 95% CI of the generic Himalaya class.

So, as you can see, no particular behavior emerges. In addition to this, the very fact that the geological map does not cover the entirety of the study area basically would leave us with two choices if we would assume that the suggestion from Rev2 is truly a requirement. Either we keep the model as is, or as you certainly know, if data is not available consistently, we would need to clip the study area to the

geological map. This would lead to an additional issue. The very fact that the high Himalaya is not available would also mean clipping away a substantial amount of landslides from our spatio-temporal dataset. Therefore, between the two choices, we prefer to keep the model as is without clipping the study area. Our model, and we reiterate this concept again, has a predictive performance, during a 10-fold spatio-temporal cross-validation of 0.9 of AUC and 0.9 or correlation coefficient for the susceptibility and the landslide area components, respectively. Therefore, adding the lithology would not really add much in terms of performance.

We mentioned this already and we are fully aware that the editor knows this very well. The geology might not be available but as we are using batches of 30m grids per 1km mapping unit, a Neural Network may be capturing the geological signature in the detailed terrain batched patterns. This point is now further discussed in the discussion section of the manuscript.

[Figure]
* * *
2. I find it very challenging also the fact, underlined by reviewer #2, that you work on a quite short time interval to be able to claim that your method is made to work on multi-temporal landslide hazard. The time component in slope stability is often decadal at the very least, even after large earthquakes, as demonstrated by various recent papers that study the co-seismic and post-seismic evolution of slope deposits across several sites worldwide after large triggering events. Could you please elaborate more on this part? It seems a very crucial one.
* * *
Thank you for pointing this out, in our case to test the developed joint model we used a short-term multi-temporal inventory, and for our prediction we used the same time interval that has been used to map the landslide (which roughly aligns with monsoon/rainy seasons, thus names pre and post monsoon). We understand that for modelling the return period of the landslides we need very long-term inventory but, in this case, we are trying to generalize our model temporally and generate temporal estimation rather than using it as a long-term forecasting tool. Since this is the first approach, we tested model's capability to do so, but in the future research with long-term data availability we could expand this work to include long-term/return-period based forecasts. This point might not have been clear; therefore, we apologize for the confusion. In the amended manuscript this point has been discussed in the discussion section.

By the way, and this is a point of reflection rather than a criticism from our side. There is no formal definition nor requirement in statistics or data-science for a specific duration in time of a space-time model. A space-time model can be defined as such as long as the variability of the response variable or variables in this case (susceptibility and landslide area density) across space and time is explained according to a set of predictors. If this is the case, then one can formally define a model as space-time. Whether this is not verified, then the definition does not hold. As our model explain a large portion of the landslide occurrence probability and of the landslide area density both in space and time, our model is a space-time one. The only limitation, and this is where we may agree with the Rev2's comment, is that the model validity is confined to the domain under consideration and as we mentioned above, it should not be used for long term prediction. This is mainly because the model has "seen" only a short time landscape evolutionary process and therefore its application over large temporal scales would not be suitable.

This being said, we have now commented on this elements of reflections and enriched the manuscript accordingly to emphasize this point.
* * *
3. I am not not acquainted with the details related to the used inventories but I trust that reviewer #2 may know them quite well. So, I am consequently worried by his assumption that they might be flawed by incompleteness (due to infrequent acquisitions in time and low resolution of images). I know that small landslide occurrences are often affected by such errors and that it is not easy to overcome limitations in inventoried data. However, I am worried about how this level of expected uncertainty may have affected your results.
* * *
Thank you for this comment. We understand the concern of Rev2 as well as the editor and we know that it might be incomplete in some extent (most the inventories are). There are two types of omission possible in this case, first temporal omission and second spatial omission. For the first case, we have limited our predictions to the same time-period as the mapped intervals (we mention this as pre-and post-monsoon in the manuscript, but the input covariates are limited to that of the mapped interval). This allowed our model to predict on the same time interval as the ones mapped by the authors of the inventory. In the spatial omission case, we believe that in a regional scale mapping there will always be some level of omission but in general if the landslides are representative and our model can learn from the samples to predict the landslides, this might affect the accuracy metrics because of faulty ground truth. However, full uncertainty propagation is out of the scope of this research, but we can discuss this point in the manuscript.

Moreover, we would like to make a point here by stating that we are introducing for the first time a space-time model capable of explaining the variability of the landslide occurrence and size patterns. Therefore, the assumed limited quality of the inventory in space or in time, can only be the basis of further tests in the future rather than being used as a mean to invalid the whole experiment. In case of a better multi-temporal inventory, the model will be tested even more, next time not in its methodological introduction in the literature but rather in its ability to work at various levels of spatial and temporal completeness. This is in the end what science is and should be. One presents a new idea or a new model, then various test can be performed in other sites, with other data before pushing a given model for a potential operational use.
* * *
4. I agree with the comment on possible overfitting issues. Some aspects of the solution you have proposed, together with the possible uncertainty in the data and lack of knowledge on key parameters

such as geology, may require a specific strategy to mitigate overfitting and minimize the generalization gap (achieving optimal capacity).

———————————————————————————————————————————————

To avoid overfitting, we have used three specific techniques, therefore the comment has limited validity here. Standard techniques to limit overfitting have been implemented as part of the modeling protocol. Specifically, we used model regularization by adding batch normalization and dropout layers to prevent model from overfitting as it is customary in deep learning architectures. Any search in any major deep learning articles is equipped with analogous techniques.

Secondly, to further avoid overfitting we used early stopping to halt the training process when model starts to overfit by using 30% of the validation data. This is also a standard techniques well adopted and represented in computer science.

Finally, we also tested our model with 50% of the unseen data to the model and all the reported metrics are based on the test set. We would like for a moment to stress that a 50% of data removal is a substantial component taken away from a spatio-temporal data. Therefore, any typical residual dependence, either in space or in time is heavily perturbed. We recall once more that all the performance metrics we report ~0.9 AUC and ~0.9 correlation coefficient for the two model components are measured on this 50%, therefore the generalization capacity of the model is more than pushed to the limit of what is conventionally done in most of landslide scientific contribution.

———————————————————————————————————————————————

Please also consider all other major and minor comments of both reviewers and please present, as usual, a point-by-point reply to them in a separate document, as well as a modified tracked version of the manuscript.

I am confident that, after covering the present issues, the manuscript will present more sound and acceptable content in terms of underlying hypothesis and analysis, and that it may become acceptable for publication on NHESS.

I thank you in advance for your submission and look forward to your submission.

———————————————————————————————————————————————

Thank you for your suggestion, we have addressed all the comments from both reviewers and provide relevant answers. We look forward to your editorial decision, confident that we have fully focused on addressing any major and minor comments raised by both the reviewers.

———————————————————————————————————————————————

**Reviewer 1 Feedback**

The authors suggest using an "Ensemble Neural Network" to holistically assess landslide hazards involving all the terms from its definition: location, time, and magnitude. They attractively demonstrate the advancement in using a data-driven model with adequate discussion, also criticizing their modelling setup, particularly the causality limitation of machine learning tools.

———————————————————————————————————————————————

Dear reviewer 1, thank you for your comments and feedback about the modelling approach.
————————————————————————————————————————

In my below comments, I pointed out a few minor issues to improve the manuscript and the confidence in the results—my comments mainly concern data and methods. A few of the suggested literature is redundant; authors should not feel obliged to involve them in the current work. I hope that the authors will benefit from my suggestions.
* * *
Dear reviewer, thank you for your comments and feedback, it surely helped us to improve our work.
* * *
Minor comments:

The authors emphasize the modelling setup's temporal aspect (advancement) in several places. However, the abstract only presents results (performance) regarding space and magnitude.

Also, the method section does not highlight how the temporal aspect and area density is assessed. I would appreciate it if those bits of the manuscript were extended. Studying the landslide legacy effect, I find the Fan 2013 paper really useful.

Huang, R., Fan, X. The landslide story. Nature Geosci 6, 325–326 (2013). https://doi.org/10.1038/ngeo1806
* * *
Dear reviewer, thank you for your feedback, we have now amended the abstract to also include temporal aspects. In our work we look at the temporal aspect only during the time that inventory is available and not beyond that because we do not have sufficient data to understand the temporal pattern of the landslides. For that reason we cannot go beyond predicting in those time frames and looing at the multiple return periods. This has been now discussed in the methods section to further clarify your point.
* * *
Do authors smooth the curvature metrics? One option could be smoothing it to average landslide size. In the meantime, total curvature is used to compute topographic amplification of seismic signals, which correlate well with the landslide activity (e.g., Maufroy et al., 2015; von Specht et al., 2019). Authors should consider experimenting with that. The second suggested article also claims that PGV is a better metric for studying coseismic landsliding than the common PGA.

Maufroy, E., Cruz-Atienza, V. M., Cotton, F., and Gaffet, S.: Frequency-scaled curvature as a proxy for topographic site-effect amplification and ground-motion variability, Bull. Seismol. Soc. Am., 105, 354–367, https://doi.org/10.1785/0120140089, 2015.

von Specht, S., Ozturk, U., Veh, G., Cotton, F., and Korup, O.: Effects of finite source rupture on landslide triggering: the 2016 Mw 7.1 Kumamoto earthquake, Solid Earth, 10, 463–486, https://doi.org/10.5194/se-10-463-2019, 2019.
* * *
Thank you for the feedback, In this study we did not smooth the curvature metrics and provide it as it is to the model which model and the given nature of the model it should be able to learn smoothing effect. Smoothing it to an average landslide size in this case is difficult because we do not know the size of the individual landslides but only the area density of landslides per 1km grid making it difficult to smooth the curvature. The total curvature would make it more interesting at the slope unit scales but

due to the 1km grid space they do not exactly match the geomorphological criterions and therefore using total curvature does not improve the model quality. For the case of PGV, we understand that PGV might be better metric but because the input data is derived from Shakemap system, and the values are derived empirically the difference in PGA and PGV is not very large, in case of directly observed or simulated data adding PGV could help by a lot. We will add this evaluation in the manuscript to make it clearer to the audience.
* * *
In a few places, authors mention limitations arising from data imbalance, e.g., line 209. Could they try sampling an equal amount of data from different classes and assessing accuracy?

There are several figures with 8 to 12 subplots. I found those figures rather uninformative. It is tough to get the main message of those figures. For example, success differences of the model over time are not apparent in Figure 8; residual differences are not evident and hard to see in Figure 9; differences between susceptibility and area density in-between and over time are not easy to recognize in Figure 10. I believe also the message of Figure 2 could be given differently.
* * *
Yes, we checked the accuracy in a balanced sample and include it in the modified text. We think it will not change the results significantly to what we already have in terms of F1 score because it is not vulnerable to imbalanced data and can represent the unbiased prediction performance. Anyway, for the shake of clarity we will include it in the amended manuscript.
* * *
Could providing correlations between subplots of Figure 11 be helpful?
* * *
Yes, we can add the correlation between those variables in the figure to include more information, which will provide the status of landslide hazard with high correlation meaning that the location with higher area density also had higher susceptibility and vice versa.
* * *
Line 30: "neglecting" à I found the statement slightly judgmental. The landslide community was primarily focused on the location aspect of landslides, as temporal landslide data was rarely available, if at all.
* * *
Thank you for your feedback, we will include your comment as a limitation and remove the word neglecting to clarify the manuscript.
* * *
Line 56: "Section ??"
* * *
Thank you for noticing, we will change it in the text.
* * *
Physics-based or Physically-based model is a better term to use? The manuscript includes both terms.
* * *
Thank you for noticing, we will change it in the text to make it uniform.
* * *
**Reviewer 2 Feedback**

After an in-depth review of the paper entitled "Space-time Landslide Hazard Modelling via Ensemble Neural Networks," it is very evident that the paper has serious flaws.
* * *
Dear Reviewer-2, Thank you for the time you dedicated to reading the manuscript. Below we will provide our responses to your comments.
* * *
1.   First of all, the paper is methodological and relies on a published multi-temporal inventory from Kincey et al. (2021). In the paper, the authors have used landslide inventory data from Nepal (Gorka post-earthquake from 2015 to 2018) and also the area density maps. The authors have used these datasets and digital elevation derivatives and rainfall and peak ground acceleration conditioning factors to produce landslide susceptibility maps and also combine the susceptibility results with area density block to give hazard output. There is clearly a methodological issue in this step as in the literature "landslide hazard refers to the probability or likelihood of a landslide occurring within a specific area and within a given period, taking into account the different factors contributing to slope instability". However, this paper clearly lacks the temporal component of hazard, which they claim they can achieve using only three years of multi-temporal inventory.
* * *
We fully agree with you that the paper is methodological advancement rather than a case study of specific slope or region. We used the data from existing sources to test the capability of our model rather than to define mitigation measures for which more careful field survey is required.

Coming back to your question on temporal component of landslide hazard, I think there is a certain misunderstanding. You state that the paper lacks the temporal component because we use three years of data. For two reasons your assumption that three years may not be sufficient does not hold, which we will clarify below.

First of all, there is no formal definition of how long should a time series be to satisfy the requirement for probabilistic modeling in time. If there is one, we would be interested in reading the source of it, so kindly provide it for our reference. This being said, we fully agree with you that if one looks into time series analyses, these make use of much longer time-windows to be carried out. However, such time series analyses, they are also carries out for single locations. In our case, as relatively short a three-year time period may be, we should keep in mind that the spatial dimension we consider covers most of the Nepalese territory. In that sense, if we trade space-for-time, the retrieved information could be fed to a space-time model. This model, if suitably built, could provide probabilistic estimates both for the spatial dimension as well as for the temporal one. As a result, the temporal dimension of the hazard can still be estimated. The only difference with a long time series requirement you imply in your comment is that the validity of the model estimates will act on a short-time framework rather than a long one. In other words, the hazard assessment our model produces is certainly valid for the three years it was built

for. We never claimed it to be valid for the next decades or centuries as it is commonly done for engineering solutions based on long return periods.

As for our second reply, it essentially expands on what explained before but focusing on the hazard definition you commented on. In fact, one can define landslide hazard models both on the basis of probabilistic and deterministic approaches. Probabilistic ones typically rely on the solution of Poisson models to provide landslide temporal frequency information such the landslide hazard likelihood for specific return periods. Such models, as you also implied in your comment, require longer term data. We also need to keep in mind that for deterministic solutions, the landslide temporal frequency is always considered 1 because we are estimating the landslide hazard with already known triggering factors. Our space-time model treats the temporal dimension deterministically. This is the main reason why we have not projected the landslide hazard in future scenarios with different return period, for instance by including climate change scenarios. This in turn is translated in a modelling approach where the temporal probability is not explicitly estimated but rather obtained from a model informed of the spatio-temporal evolution of landslide occurrences and planimetric characteristics.

We hope to have provided sufficient evidence on why our modeling protocol is not flawed but simply framed in a different structure as compared to more standard alternatives. This being said, your comment made us realize that all the discussion provided here was not expressed clearly enough in the manuscript, or at least, it made us realize that more effort should be put into providing a clearer justification for our choices and assumptions. In the revised version of the manuscript, we have added a detailed description of the notion we introduced above as well discussed the temporal aspects of the model. Overall, we would like to thank you for your comment, as we believe it will indeed improve the text and the readability for the NHESS readership.
* * *
2. In any landslide hazard modelling paper, geological aspects cannot be ignored to model landslide hazard. However, this paper lacks a section discussing the geological assumptions made during the study. Understanding the geological context is crucial for any Earth science study, more so for hazard modelling. There are several other data bias issues and model selection uncertainties. Still, the authors are repetitively putting emphasis on the novelty aspect as mentioned in lines 11-14 and again in lines 23-25, without substantiating their claims with solid arguments or evidence.
* * *
Thank you for this insightful comment, indeed we did not include a section in the geological assumptions and aspects in the area because as you already mentioned our work is directed towards methodological advancements rather than focusing on a detailed case study. We would like to remind here that space-time modeling for landslide hazard estimation across large geographic scales has very few contributions, which is where our interest and efforts have been directed to. As for the role of geology, it is equally important to realize that for Nepal and specifically for the entirety of the area under consideration, the availability of detailed lithological maps is extremely limited, if not absent. We are not stating that Nepal does not have relevant lithological information. However, this is only valid for specific sectors. The area where we designed our experiment and modeling protocol is lithologically described into four classes. To provide evidence of our statement, we would refer the anonymous reviewer to the geological map available at the following link:

https://certmapper.cr.usgs.gov/data/apps/world-maps/

Due to the reviewer expertise on data-driven modeling, we are sure you would understand that a subdivision of an area that basically almost covers the whole country of Nepal in just 4 lithological

classes would not support any realistic geological assumption. You may wonder if other sources of lithological information are available for the whole study area under consideration. We have looked into this and found two more sources.

These can be found at the two following links and below we will explain why there are equally useless.

Link1: https://www.data.gov.uk/dataset/460872e8-7a77-45c6-90c6-9b979fcae0d2/simplified-geological-map-of-central-eastern-nepal-nerc-grant-ne-l002582-1

Link2: (PDF) Numerical Modeling for Support System Design of Headrace tunnel of Rahughat Hydroelectric Project (researchgate.net)

Both these sources, as you will see from the second link in Figure 1-1, do provide a slightly better geological characterization from the spatial perspective. In fact, the number of classes are seven. Here we should remark once more that seven classes are still a very small number compared to the extent of the study area, leaving any geological consideration unsubstantiated in any case. But, let us assume that they are enough. The issue is that they only refer to geological formations, which makes it impossible to interpret any spatio-temporal dependence with respect to landslide occurrences and relative areal densities. For instance, how would one address your request of providing a sound geological explanation if the class is "Lesser Himalayan Zone" or "Higher Himalayan Zone". It goes without saying that any explanation will end up becoming a speculation, for which the reviewer could be equally critical. Moreover, as we also referred to the editor, the best map we could find also could not really separate the distribution of landslides to be statistically significant. There is a 1:50,000 map available from mines and geology department of Nepal but it only covers very small portion (https://dmgnepal.gov.np/en/resources/geological-maps-150000-4749) of the study area and is only available as printed map/PDF and not as a digital data so we did not spend a lot of time digitizing it without any added value.

This is to say that the comment from the reviewer does make sense from a pure theoretical perspective. However, its practical feasibility is much less reasonable when considering the data availability across the whole study area.

This is our most generic answer but another element to be addressed here is to ask ourselves whether one would actually need such thematic information. Our space-time model offers outstanding performance both in the susceptibility component as well as in the area density one. So, what would add the use of lithology? If we would really add it and the model would suddenly predict 100% of the landslide occurrence location as well as their planimetric extents, one could say that the model would suddenly be unreliable because it is impossible to predict everything correctly. This is for us to explain that modeling requests should also follow a feasibility criterion, which is not the case here upon consideration of data availability and also on why such information should be useful at all. Most likely, a very complex model as the one we implemented here is capturing micro-to-marco scale geological effects through the use of terrain characteristics. We should remember here that, yes, our mapping unit of choice is a 1x1 km2 lattice. However, the information is passed to the neural network as an nested partition at approximately 30m resolution. For this reason, the model may intrinsically learn that 30m pixels at 90 degrees could only be possible if the material is rocky in nature and that much gentler slopes may be characteristics of softer or unconsolidated materials.

Having provided extensive evidence of why the concerns raised by the reviewer do not apply to our case from a pure modeling perspective, we must admit that while re-reading the document, we also realized that the description of the geological context at large could have been largely improved. For this reason, in the revised manuscript have added a section to describe the geological context of the study area and the limitations we faced.

3. The introduction section of the paper is very poor, with one paragraph related to physically based modelling and another with statistical models for landslide susceptibility. In recent decades, several high-quality papers have been published related to the use of neural networks for landslide susceptibility modelling and for space-time modelling such as

"Montrasio, L., Valentino, R., Corina, A. et al. A prototype system for space–time assessment of rainfall-induced shallow landslides in Italy. Nat Hazards 74, 1263–1290 (2014) https://doi.org/10.1007/s11069-014-1239-8.

"Nocentini, Nicola, et al. "Towards landslide space-time forecasting through machine learning: the influence of rainfall parameters and model setting." Frontiers in Earth Science 11 (2023): 1152130."

"Grelle, G., Soriano, M., Revellino, P. et al. Space–time prediction of rainfall-induced shallow landslides through a combined probabilistic/deterministic approach, optimized for initial water table conditions. Bull Eng Geol Environ 73, 877–890 (2014). https://doi.org/10.1007/s10064-013-0546-8".

"Catani, Filippo, Veronica Tofani, and Daniela Lagomarsino. "Spatial patterns of landslide dimension: a tool for magnitude mapping." Geomorphology 273 (2016): 361-373."

Thank you for your comment. We do agree that the literature review can be expanded and this is what we plan in the revised version of the manuscript, including the references you suggested. We would like to stress that in the original version, we tried to provide as concise and clear introduction as possible to the audiences, doing so, we might have missed some of the literature, we will further expand our manuscript to include the referred literatures as well.

4. The results of this paper are the product of incomplete datasets, and it is evident from the paper by Kincey et al. (2021), that for an area of about 25000 km² from 2014 to 2018, only two experts manually digitized the landslide inventories every six months, with less frequency in 2017-2018. They used coarse-resolution imagery from Landsat and Sentinel-2 for manual visual interpretation of landslides. Due to the use of coarser resolution imagery, many small landslides along road networks and even in areas closer to built-up areas were missing. It also resulted in spatial bias in landslide area density. The main concern is that the authors used the datasets from Kincey et al. (2021) without any new consideration and quality check, trying to model landslide hazard on a dataset not originally intended for the direct usability of creating a landslide hazard map. This leads to model uncertainty regarding training, prediction, and especially validation.

We generally agree with you for this comment. We are fully aware that the multi-temporal inventory is developed from relatively coarse resolution dataset (even though, 10 m sentinel 2 can be classified as high resolution in every sense of its term) and does not contain the landslides that are likely triggered by anthropogenic factors such as near road and near built-up areas. However, it is also important to stress that the potential bias you refer to could theoretically make its way into the model

if captured by covariates that are linked to anthropogenic effects. Specifically for this reason, we have not explicitly included any anthropogenic conditioning factors such as road network and built-up area in our model. In other words, by only using natural predictors such as terrain and meteorological characteristics, our model would not reflect inventory biases in its final estimates. This topic is

extensively discusses and made very clear in a number of contributions made by Stefan Steger. Specifically, Steger et al. (2016) or (2021) precisely explain how inventory biases can affect model results if captured by bias-related covariates, which we avoided. Therefore, your concerns are generally sound but not in the specific version of the space-time model we implemented.

As for the comment on visual interpretation, we think that manual and visual mapping of landslides is in fact the most appropriate way to map the landslides after field visit (which is not possible for such a vast region). Even though the machine learning based approaches are fast, they are prone to make more mistakes than visual approaches and we think this inventory in that sense is good. The dataset provided from Kincey et al. (2021) is in a gridded structure as an area density of the landslides rather than the actual polygons in which case we could look at the frequency area distribution and perform quality check. However, due to lack of the actual landslide polygons we could not perform this test. Moreover, this is the first landslide inventory data that covers such a large area of landslide evolution which is enough to train a deep learning model meaningfully, which makes it suitable for our application and test case.

Regarding the model uncertainty, yes because it is a regional scale model, it has many uncertainties, and they are inherit problems with all of the scientific methods. In this case, main uncertainties come from the inventory and predisposing factors and we would like to stress that uncertainty propagation should be further studied in terms of landslide hazard modelling but this is out of scope for this manuscript.

Having responded to all your comments, it is important for us to mention that if the concerns you raised here would become a requirement for any article dealing with prediction of landslides, then half of the literature would not exists. For instance, specifically for earthquake-induced landslides, it is common practice using landslide inventories shared by the teams responsible for the mapping, even beyond the original author list. In other words, inventories are used for testing certain hypotheses, even if they have certain limitations. For instance, let us take the example of an author you mentioned above among the suggested references, Prof. Filippo Catani. He has recently co-authored a number of articles (e.g., Loche et al., 2022 or Meena et al., 2023) relying on inventories generated by others. Let us look into "HR-GLDD: a globally distributed dataset using generalized deep learning (DL) for rapid landslide mapping on high-resolution (HR) satellite imagery". In their work, the authors make use of 13 inventories from very different source to train a global automated landslide mapping tool based on Neural Networks. Among these 13 inventories there will certainly be some degree of influence or bias brought by the original resolution of the satellite images used for mapping or even by the different group members among those responsible for the respective mapping procedure. However, does this affect the quality of their methodological contribution? Not at all. Yes, some bias and uncertainties are unfortunately inevitable, but the best one can to is to not introduce predictors that could propagate the potential bias across the whole modeling pipeline, which is what we did. We do not see any other clear bias removal procedure to be included. All this is to say that we did our best, and to address your concerns, in the revised manuscript we will comment on these elements in the discussions, as per your comment when you refer to "considerations and quality check".
* * *
5.      The paper is vague about the exact architecture of the Ensemble Neural Network (ENN). Information about layers, nodes, and activation functions is missing. There is a complete absence of discussion on hyperparameter tuning, which is critical for the performance of deep learning models.

The paper doesn't provide details on the training procedures, such as batch sizes, learning rates, or optimization algorithms used, impacting the model's reproducibility. While claims are made about the model's satisfactory performance, there is no elaboration on how this was evaluated, such as specific metrics or comparative baselines. The paper doesn't discuss the computational resources required for training and implementing the model, which is vital information for potential users. The paper lacks explicit discussion about the assumptions behind the models used. This makes it difficult to assess the reliability and applicability of the results. The section about selection of mapping is unclear and does not satisfy the reasoning given to modify the mapping units; it seems it was deliberately done to fit their model needs to achieve better results.
* * *
This is the comment where we mostly disagree with you and we will provide clear evidence on why this is not the case. The model architecture is well defined in the section 5.1 with details of each layer and how they are connected. In addition to this, Figures 5 and 6 include details of each layers and their shape as well as connection parameters. The information of layers in Figure 6 refers to each Resnet block, which includes convolution 2d layers followed by batch normalization ReLU activation function and max pooling layers. This is all written in the text and we ask the reviewer as well as the handling editor to confirm directly our statement in the original version of the submitted manuscript.

As for the comment on the hyperparameters, while reviewing the text, we understood your comment and would like to acknowledge here its validity. To answer your comment, hyperparameters are tuned using the keras tuner. We have tuned the depth of the network, learning rate and the batch size. The width of the model is not tuned because we have derived our model from an well-established approach. The reason why we agree with you here is that we realized this information was not included in the current manuscript and we will certainly include a better description in the revised manuscript.

This is to say that we are trying to find a middle-ground between your comments and our starting point. However, even if we agree to some extent on your hyperparameter comment, it is also true that there is a dedicated section on section 5.2 Experimental setup (and again, we refer both the reviewer and the handling editor to the original version of the manuscript).There we explicitly mention learning rate, batch size, optimization algorithms. For reproducibility, we have even shared the model and code since the very beginning of the submission process. There, our choices are very transparent and accessible to anyone.

With respect the comment on model's evaluation metrics, we struggle again to agree with you and same as above, we will provide evidence below.

In fact, section 5.3 Performance metrics offers a clear overview on the metrics we used. These are expressed in terms of AUC for the classification part of the ensemble model and as a Pearson correlation for the regression element. As for your comment on the requirement for a baseline, this is not possible. There is simply no space-time data-driven model in the literature equipped with a dual component for susceptibility and area density estimation. The only available model in the literature pertains to the susceptibility notion, for which a 0.9 AUC corresponds to outstanding prediction skills. Also, here we recall that this value corresponds to the prediction skill of the model, as it is estimated on a subset of the spatio-temporal data which was never "shown" to the neural network architecture. As for the regression component, there is no analogous experiment in the literature.

Regarding computational resources, it is a purely technical requirement and using the minibatches and dynamic data loading most of the machine learning related infrastructure can easily run the deep learning based model. Specifically, we have used 32 AMD Ryzen Threadripper PRO virtual CPU, in a machine with 160 GB RAM, and NVIDIA RTX A4000 GPU. However, this is a shared resource on a computational infrastructure of the University of Twente and we did not fully use the full capability of the machine. Most of the training was done with approximately a 30-40% load.

As for your comment about the mapping unit, we do not understand what you refer to. We never modified the mapping unit at any point in the manuscript. Our mapping unit of choice is the same as the resolution of the data provided by Kincey et al. (2021), something we chose for reason of consistency. If you are referring to the 4km x 4km scheme illustrated in the manuscript, this is the image patching technique required to create small patches of the dataset, something extremely common in any convolutional neural network models. The reason for which this is commonly done has to do with the number of samples which would otherwise just be 1 if we input entirety of data. To be fair, as we went through the manuscript, we realized this could have been explained better. During the revision process, this is something we will certainly address and clarify to the NHESS readership.

———————————————————————————————————————————————————

6.   The paper details a neural network model with 23,556,931 trainable parameters but does not discuss how overfitting is mitigated. This is a significant concern, especially if the dataset is insufficient to justify such complexity. There is a lack of information on how the hyperparameters for the Adam optimizer and the learning rate were selected. This absence of methodological detail hampers the paper's replicability. The paper glosses over crucial data preprocessing steps and how imbalanced data is handled. Given the nature of landslide data, this could be a major issue affecting the model's performance. There is no discussion on model validation techniques like cross-validation, raising questions about the model's generalizability. The choice of a 1km x 1km grid for spatial analysis is unjustified. Given that landslides are highly local and temporal phenomena, failing to account for these could result in a model with limited applicability. The paper is limited in the range of environmental factors considered, focusing only on earthquake and rainfall intensities. This narrow scope risks omitting crucial predictors of landslides. The approach of training two separate components and then combining them is unconventional and could introduce errors or biases, none of which are discussed. The text is quite complex and convoluted, making it difficult for readers to follow the methodology and the presented arguments.

———————————————————————————————————————————————————

Same as before, if this level of question is the point of discussion or criticism, then we have shared absolutely everything in a transparent manner in github. We just did not think of providing all technical nuances in the main text because the journal is still an applied one rather than belonging to the category of information science. To address your question we will explain below those details and stress beforehand our availability to introduce this information in the text during the revision.

The overfitting is mitigated by two major approaches, firstly by the use of batch normalization and dropout layers where the model itself is optimized to prevent overfitting and secondly, by the use of early stopping. The early stopping feature in the model training process checks the validation performance at each step of the model and if the model starts to overfit and validation loss starts to increase (thereby implying a decrease in validation accuracy) significantly the model would stop

training automatically. These aspects have been mentioned in the text as the reviewer and the editor can check in the original version of the manuscript. However, we realized these are not been explicitly mentioned to address overfitting issues. We thought this was something quite straightforward for machine learning users, and we apologize if we have created any confusion. We will amend the manuscript accordingly during the revision process.

For the learning rate, we used to tune the learning rate from 1e-4 to 1e-1 range as an initial value and then decayed the values and selected the best among them (best in the sense that it converge fast and without fluctuations). Same as before, in the amended manuscript we will add further details on hyperparameter selection for more clearer understanding.

In case of data imbalance and data pre-processing, we have explicitly mentioned about the tackling of data imbalance in the section 5.2 with specific use of Focal Traversky loss and use of logarithmic training process for both components in the model. Furthermore, to cross validate the model's output we have used metrics that are not/less affected by data imbalance (such as F1 score) and focuses on positive samples.

As we mentioned already in a pervious comment, the 1x1km grid is not a deliberate choice but it is a choice due to the availability of landslide inventory as an area density in 1x1 km grid from Kincey et al (2021), which we kept to build our model accordingly. This shape can be modified depending on the inventory quality and dataset.

The focus on earthquake and rainfall is because they are the crucial triggering factors, we deliberately did not include the anthropogenic factors because of the quality of inventory. Those predictors in our opinion are the most commonly used and significant conditioning factors and for generalization and application at other locations readers are encouraged to select the input data as per the availability and geomorphological as well as geological context.

Actually ensemble approach and combined training as well as separate training of the two components of a same output is conventional approach in computer sciences and many other fields but not in the landslide science because there has not been any model working in this direction. Using separate models and combining them in the later stage reduces the error because if both models separately agree on the location of landslide (given the size is conditional to occurrence), this should indicate that they both provide reasonably good outcome. This ensemble approach will be further clarified in the revised manuscript to include further details.
* * *
Overall, this methodological paper is not suitable to be published in its current form and low level of scientific quality in a high-impact journal such as Natural Hazards Earth System Sciences.
* * *
Thank you for your comment and feedback overall, we will leave it up to the editor to decide whether the manuscript is with enough methodological novelty and scientific advancement to publish on NHESS or not.